# A daily reconstructed chlorophyll-a dataset in the South China Sea from MODIS using OI-SwinUnet

Haibin Ye[1,2], Chaoyu Yang[3,4], Yuan Dong[1,2], Shilin Tang[1,2], Chuqun Chen[1,2]

[1]State Key Laboratory of Tropical Oceanography & Guangdong Key Laboratory of Ocean Remote Sensing, South China Sea Institute of Oceanology, Chinese Academy of Sciences, Guangzhou, China
[2]Southern Marine Science and Engineering Guangdong Laboratory (Guangzhou), Guangzhou, China
[3]South China Sea Marine Forecast and Hazard Mitigation Center, Ministry of Natural Resource, Guangzhou, China
[4]Key Laboratory of Marine Environment Survey Technology and Application, Ministry of Natural Resource, Guangzhou, China

*Correspondence to:* Shilin Tang (sltang@scsio.ac.cn)

**Abstract.** Satellite remote sensing of sea surface chlorophyll products sometimes yields a significant amount of sporadic missing data due to various variables, such as weather conditions and operational failures of satellite sensors. The limited nature of satellite observation data impedes the utilization of satellite data in the domain of marine research. Hence, it is highly important to investigate techniques for reconstructing satellite remote sensing data to obtain spatially and temporally uninterrupted and comprehensive data within the desired area. This approach will expand the potential applications of remote sensing data and enhance the efficiency of data usage. To address this series of problems, based on the demand for research on the ecological effects of multiscale dynamic processes in the South China Sea, this paper combines the advantages of the optimal interpolation (OI) method and SwinUnet and successfully develops a deep learning model based on the expected variance in data anomalies, called OI-SwinUnet. The OI-SwinUnet method was used to reconstruct the MODIS chlorophyll-a concentration products of the South China Sea from 2013 to 2017. When comparing the performances of the DINEOF, OI, and Unet approaches, it is evident that the OI-SwinUnet algorithm outperforms the other algorithms in terms of reconstruction. We conduct a reconstruction experiment using different artificial missing patterns to assess the resilience of OI-SwinUnet. Ultimately, the reconstructed dataset was utilized to examine the seasonal variations and geographical distribution of chlorophyll-a concentrations in various regions of the South China Sea. Additionally, the impact of the plume front on the dispersion of phytoplankton in upwelling areas was assessed. The potential use of reconstructed

products to investigate the process by which individual mesoscale eddies affect sea surface chlorophyll is also examined.

## 1. Introduction

Chlorophyll-a is the primary pigment involved in photosynthesis in phytoplankton. Its concentration serves as a crucial metric for assessing the density of phytoplankton, including algae

(Vantrepotte and Mélin, 2011; Ye et al., 2020). Conventional techniques for measuring chlorophyll-a concentrations use ship surveys and buoy observations. Sampling using these methods is not only expensive and time-consuming but also challenging to implement for monitoring on a wide scale and over extended periods. Consequently, this approach fails to accurately capture real-time fluctuations in chlorophyll-a concentrations over time (Bierman et al., 2011). Due to advancements in satellite remote

sensing technology, remote sensing has emerged as the primary method for acquiring ocean color data (Gregg, 2007). Satellite-based ocean color remote sensing allows convenient retrieval of sea surface chlorophyll-a concentration data with high temporal and spatial resolutions. Nevertheless, the majority of ocean color satellites follow a polar orbit, resulting in limited data coverage due to the gaps between scanning flights. In addition, the water column's limited capacity to reflect signals can be influenced by

certain meteorological factors, such as cloud cover, solar flares, and dense aerosols. These factors can impact the accuracy of sensor sampling, leading to missing data (Wang and Shi, 2006; Liu and Wang, 2018). The absence of these data will significantly restrict its future utilization. Data reconstruction is a technique that can bridge gaps in data, expand the range of ocean color data, and ensure that the data are continuous throughout time and place. The reconstructed ocean color data not only maintain the

temporal consistency of the data but also expand the spatially valid range of the data, enabling a more accurate representation of the continuous distribution and variation in sea surface chlorophyll-a concentration in the ocean over time and space (Barrot, 2010).

Two frequently employed reconstruction approaches in the field are optimum interpolation (OI) (Reynolds and Smith, 1994; Kako and Isobe and Kubota, 2011) and data interpolating empirical

orthogonal functions (DINEOF) (Beckers and Rixen, 2003). The OI algorithm leverages the conservative nature of marine elements and takes into account the spatial distribution characteristics of each element. It interpolates the unevenly distributed data to the corresponding grid points, resulting in

an optimal estimation. This algorithm increases the coverage area and data density, allowing for the simultaneous use of observation data with varying error characteristics. It effectively addresses the issue of sparse spatial distribution of marine data. The optimal interpolation method has gained global recognition since the 1980s and has been adopted by the U.S. National Meteorological Center (NMC) and the European Centre for Medium-Range Weather Forecasts (ECMWF) for assimilation analysis and numerical prediction (Shaw, 1986). The method is extensively employed in the marine domain to reconstruct historical datasets of sea surface temperature (SST), in situ measurements, and sea level anomaly (SLA) datasets. Currently, it is the most often used data assimilation method in the field of marine meteorology. The assumption made by "OI" is that the datasets are independent in terms of space and time. However, it fails to adequately consider the spatial and temporal correlation of the data. The suboptimal computational efficiency of the optimal interpolation approach is also a constraining factor in its implementation.

DINEOF is a data reconstruction technique that relies on the use of Empirical Orthogonal Function (EOF). It possesses the benefit of internal adaptive correlation without requiring any predetermined values for variables. The cross-correction set is implemented to facilitate the optimal reduction of truncation and estimation errors when constructing the EOF by accounting for default values. This method not only addresses missing data and eliminates noise from the data image, but also produces a dynamically adjusted image that accurately represents the overall condition of the data and its temporal development trend. This is achieved by utilizing the most significant modes obtained through optimal truncation (Alvera-Azarate and Barth and Rixen, 2005). Due to the fact that the initial modes in the DINEOF method, which are derived from the entire target dataset decomposed by EOF, represent changes that occur over a period of more than six months, the reconstruction of multi-year time scale large data volume satellite remote sensing datasets using the DINEOF method focuses primarily on capturing temporal and spatial large-scale information. It disregards the small-scale information from a few local observation points. Therefore, using the interpolated target ocean dataset with missing measurements generated by the DINEOF method is not suitable for studying temporal small-scale processes, such as local weather-scale phenomena.

The use of deep learning techniques, specifically the CNN-based Unet model, has been proposed by researchers for the purpose of reconstructing chlorophyll-a products derived from Moderate Resolution Imaging Spectroradiometer (MODIS)/Aqua (Ye et al., 2023). Unet is a compact

convolutional neural network architecture that includes an encoder-decoder framework, which involves downsampling and upsampling operations. Additionally, Unet incorporates Attention Gates (AGs) inside its network structure. By training Unet with AGs, the background regions in the image are suppressed while the salient features in the data-missing regions are highlighted. This leads to an improvement in the sensitivity of the model and the accuracy of reconstruction. This model has demonstrated favorable outcomes in the region of the Pearl River Estuary, which is characterized by highly turbid waters. However, the applicability of the CNN model to a larger expanse, such as the South China Sea, has raised concerns. This is primarily due to the intricate nature of the physical oceanic processes in the South China Sea. Furthermore, the limited capacity of the CNN model may hinder its ability to effectively capture the comprehensive characteristics of the entire South China Sea.

In recent years, Google, Inc., has presented the Transformer architecture as an alternative to the CNNs and RNNs. This architecture is exclusively based on the self-attention mechanism and feed-forward neural network. The self-attention mechanism has superior parallelism capabilities and effectively addresses the challenge of long-range dependencies (Vaswni et al., 2017). The hierarchical transformer design, known as the Swin Transformer, was proposed by researchers at Microsoft Research Asia in the field of computer vision. This architecture computes its representation by shifting windows. The utilization of a shifted windowing in the Transformer enables interactions between neighboring windows, resulting in a significant reduction in computational complexity. Moreover, the hierarchical design resembles the hierarchical construction frequently employed in convolutional neural networks. Consequently, this technique can be effectively applied to various tasks, including but not limited to, image classification, image segmentation, and target detection (Liu et al., 2021). SwinUnet was developed by researchers as an extension of the Swin Transformer. It utilizes the Swin Transformer as its fundamental module and is composed of an encoder, bottleneck, decoder, and skip-connection. SwinUnet establishes a connection between shallow and deep features via skip connections akin to Unet,. This approach effectively mitigates the loss of spatial information caused by downsampling, resulting in notable improvements in accuracy, robustness, and generalizability (Cao et al., 2023).

This paper aims to address the existing challenges and research gaps in traditional reconstruction methods and CNN-based reconstruction models. It focuses on studying the mechanism of chlorophyll in multi-scale spatio-temporal changes in the South China Sea (SCS), including weather-scale. To

achieve effective filling of missing data in remotely sensed data products, we proposes a novel approach called the OI-SwinUnet method. This method combines the techniques of optimal interpolation (OI) and SwinUnet, and utilizes a multi-scale optimal interpolation, quadratic revision of transformer-based U-type coding and decoding network.

## 2. Materials and Methods

### 2.1 MODIS Imaging

The Level-1A data for MODIS/Aqua and MODIS/Terra were acquired from the Ocean Water Color Archive, which is maintained by the National Aeronautics and Space Administration (NASA). The remotely sensed data were preprocessed using the SeaWiFS data analysis system (SeaDAS 8.3.10). The MUMM algorithm was employed for atmospheric correction (Ruddick and Ovidio and Rijkeboer, 2000), while the CI algorithm was utilized for the retrieval of chlorophyll-a concentrations (Hu and Lee and Franz, 2012; Hu et al., 2019). Subsequently, the daily data were processed to generate Level-3 Standard Mapped Images (SMIs) with a spatial resolution of 1 km.

### 2.2 Methods

#### 2.2.1 General Framework

The primary aim of this research is to design a comprehensive deep learning framework that can effectively address the issue of missing observed chlorophyll-a concentration data. This framework will be trained using satellite-observed chlorophyll-a concentration data and will incorporate in situ observations to improve the accuracy of the reconstructed data. To accomplish this objective, we utilize the optimum interpolation approach and SwinUnet. A schematic representation of the suggested framework is depicted in Fig. 1. The objective of our study is to employ optimal interpolation for integrating satellite and in situ observations by leveraging the spatial domain information of the observations. Additionally, we aim to utilize SwinUnet to conduct multiscale feature learning on remotely sensed observational time-series data across a vast geographical area and extensive time series. Ultimately, the goal is to estimate the marginal distributions for all locations with missing values.

In practice, the remotely sensed chlorophyll-a concentrations of MODIS/Aqua and MODIS/Terra

in the South China Sea from 2013-2017 were reconstructed. This reconstruction was performed to create a daily multisatellite merged product that would provide complete coverage of the product in both temporal and spatial dimensions. The dataset consists of 1826 time series, with each image having a pixel matrix size of 2240×2240. The spatial resolution of the images is 1 km, and they cover a spatial extent ranging from approximately 0-25°N and 100-125°E.

Initially, the level-3 chlorophyll-a concentration products derived from the two MODIS sensors, were sequentially included in the OI module. This process yielded an OI merged product for determining the chlorophyll-a concentration. The OI merged product serves as a background for producing input variables for the SwinUnet module.

Next, the log-transformed satellite observation is subtracted from the log-transformed OI
background to yield the disparity between the two. To maintain the reliability of the reconstructed data, researchers have proposed the establishment of a missing data rate threshold. This threshold would exclude data with a missing rate over the specified threshold from being utilized for training and validation purposes (Barth et al., 2020). This work establishes a missing data rate threshold of 0.6, taking into account the specific characteristics of the remote sensing products in the study area. As a
result, a total of 422 time-series datasets were successfully retrieved. The training dataset consisted of 379 randomly selected time-series data points, while the remaining 43 time-series data points were used as the validation dataset. The purpose of the validation dataset was to assess the model's performance by comparing it with the reconstruction results. Notably, the validation dataset was not utilized during the training process of the model.

This study introduces a 3D tensor as the input for SwinUnet, with a specific size of $H \times W \times C$. The parameters $H$ and $W$ represent the dimensions of the image, specifically its height and width, whereas the parameter $C$ denotes the number of channels. This document presents the variables of the first to the sixth channels, listed in the following order:

1)    Disparities in chlorophyll-a concentration, adjusted by the reciprocal of the error variance
($ANO_T/\sigma_T^2$), while accounting for missing data using zero filling.

2)    The reciprocal of the variance of errors ($1/\sigma_T^2$) is calculated, with missing data imputed as zeros.

3)    The longitude data were adjusted to a range of -1 to 1.

4)    The latitude data were adjusted to a range of -1 to 1.

5)    The temporal data were linearly scaled according to the cosine function.

6)    The temporal data were linearly scaled according to the sine function.

The variables in the remaining 60 channels exhibited disparities in chlorophyll-a concentrations, which were adjusted by the reciprocal of the error variance and the reciprocal of the variance of errors. These variables are calculated for two time periods: from the pre-15 day to the pre-1 day and from the post-1 day to the post-15 day. In cases where data are missing, the term is replaced with zeros. There

are a large number of mesoscale processes such as eddies and fronts in the South China Sea, and their time scales range from a few days to several months. Theoretically, the longer the duration of the input variables, the more useful features the model can learn from the training. However, it is not possible to maximize the length of the input variables without any limitations, and the choice of fifteen days for the model inputs is a combination of many factors. Such a choice covers a complete mesoscale process

as much as possible, while taking into account the computational efficiency.

SwinUnet yields a 3-dimensional tensor with dimensions $H \times W \times 2$ as its output. The first layer ($H \times W$) represents the scaled disparity in the chlorophyll-a concentration adjusted by the inverse of the expected error variance ($ANO_P/\sigma_P^2$). The second layer ($H \times W$) represents the logarithm of the inverse of the expected error variance $\log(1/\sigma_P^2)$. The KL divergence was selected as the loss function

and computed using the following formula:

$$Loss = KLDiv(log\_softmax(ANO_T \times Mask), softmax(ANO_P \times Mask)) \qquad (1)$$

The equation for calculating $Loss$ considers the true value ($ANO_T$), which is obtained by subtracting the OI background from the satellite-observed values. $ANO_P$ represents the model-predicted value, which corresponds to the predicted discrepancy. The missing value mask (Mask) is used to ensure that only pixels with valid values are included in the $Loss$ calculation. The occurrence

of negative loss values can be mitigated by employing the $log\_softmax$ function and the $softmax$ function on the true and predicted values, respectively. The reconstructed chlorophyll-a concentration was obtained by adding the model-predicted value to the OI background value. The formula is expressed in the following manner:

$$CHL_{rec} = CHL_{OI} + ANO_P \qquad (2)$$

The training process was expedited by the utilization of a graphics processing unit (GPU), while

the selection of the optimizer was based on the Adam algorithm (Kingma and Ba, 2015). The default

parameter values were employed, including a learning rate of 0.001, beta values of 0.9 and 0.999, and an epsilon value of 1e-8. The decision to set the batch size of the training dataset to 1 was made based on considerations of the GPU memory and the size of the input tensor.

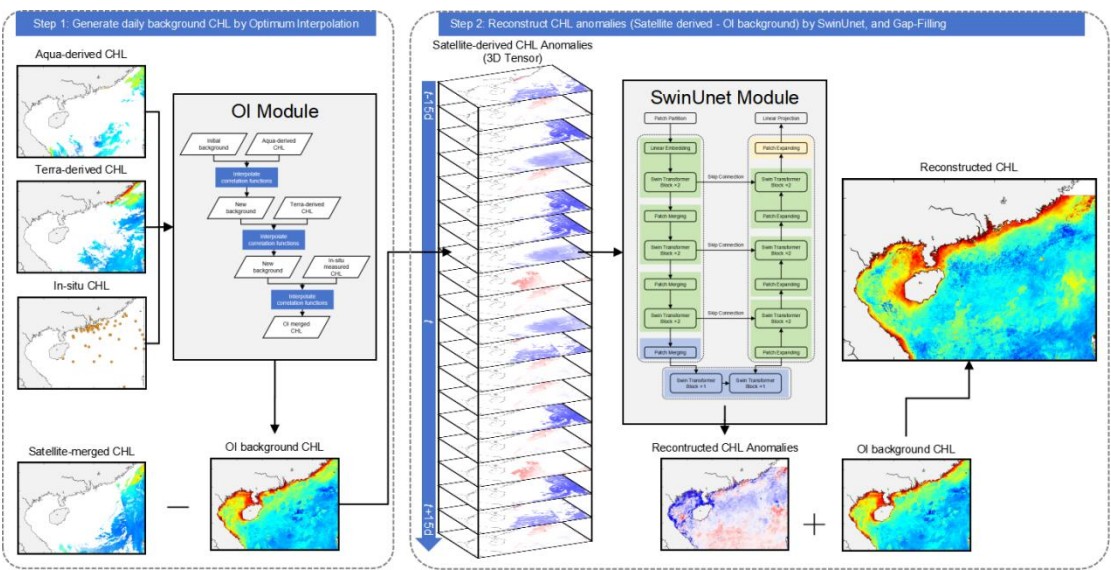

**Figure 1: Overview of the structure of the OI-SwinUnet framework**

### 2.2.2  OI Module

The optimum interpolation is a method of analysis that aims to minimize the variance of the answer analytically. This is done by assuming that the background, observed, and analytical values are all unbiased estimates (Reynolds and Smith, 1994; Kako and Isobe and Kubota, 2011). The process of optimum interpolation involves determining the analyzed value at each spatial grid point by considering the original valuation of the grid point along with a revised value. This revised value is

calculated based on the deviation of the known observations from the initial valuation of the model at $N$ grid points within a specified range, as represented by the following equation:

$$W = BH^T(R + HBH^T)^{-1} \tag{3}$$

The matrix $B$ represents the initial estimation error covariance, while the matrix $R$ represents the observation error covariance. The computation of the initial estimate error covariance matrix can be computationally demanding. In practice, it is common to utilize a matrix decomposition technique to

approximate the initial estimation error covariance matrix without taking into account the equilibrium operator, denoted by $B = D^{\frac{1}{2}}\rho D^{\frac{1}{2}}$ (Wang et al., 2014). The diagonal matrix $D$ is composed of the

variance of the initial estimate field. The matrix $\rho$ represents the correlation between the elements of the initial estimate field. Similarly, the observation error covariance matrix $R$ undergoes the same treatment. The weight function $W$ can be computed by estimating the initial estimate error covariance matrix and the observation error covariance matrix, which allows for the determination of the gridded analysis values.

The background field within the optimum interpolation refers to the first assessment of the observations. The data of the background field are then adjusted based on the sensor's observations and the associated weighting function. This study used the average chlorophyll-a concentration from the corresponding seasonal months in the two preceding years as the background field for data merging. The implementation of optimum interpolation involves utilizing the optimum interpolation algorithm model with single-sensor data and background field data as inputs. The resulting optimum interpolation merge data are then employed as the updated background field. This process is repeated iteratively by substituting another single-sensor dataset into the model, ultimately yielding the merged multisensor optimum interpolation dataset (Fig. 2).

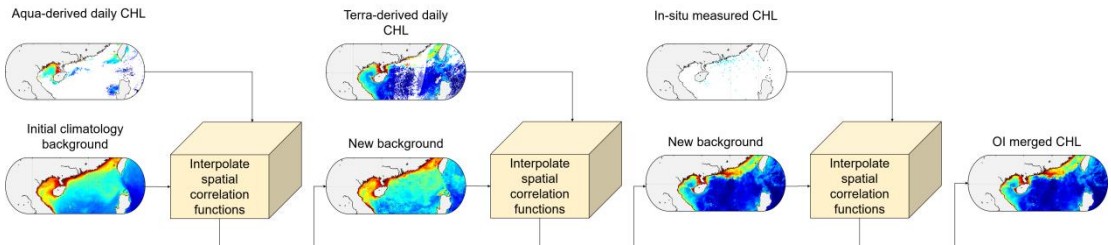

**Figure 2: The flowchart depicting the process of data merging using optimum interpolation**

### 2.2.3 SwinUnet Module

The Unet architecture serves as the fundamental framework for SwinUnet. The model comprises four primary components, namely, the encoder, decoder, bottleneck, and skip connections (Fig. 3). Given that the original SwinUnet model requires 3 channels of input data, we encountered disparity because our preprocessed data comprised 66 channels. To address this discrepancy, we introduce an additional convolutional layer prior to the patch partition layer. This new layer serves the purpose of transforming the data from its original 66 channels to the required 3 channels. In order to transform the image into an embedding sequence, we divide the entire input tensor into patches of size $4 \times 4$ that do

not overlap. These patches are then flattened in the direction of the channels. By employing this partitioning technique, the dimensions of the feature map transform from $[H, W, 3]$ to $[H/4, W/4, 48]$. Next, the linear embedding layer linearly transforms the feature dimension of each pixel from 48 to $C$. This results in a change in the shape of the feature map from $[H/4, W/4, 48]$ to $[H/4, W/4, C]$.

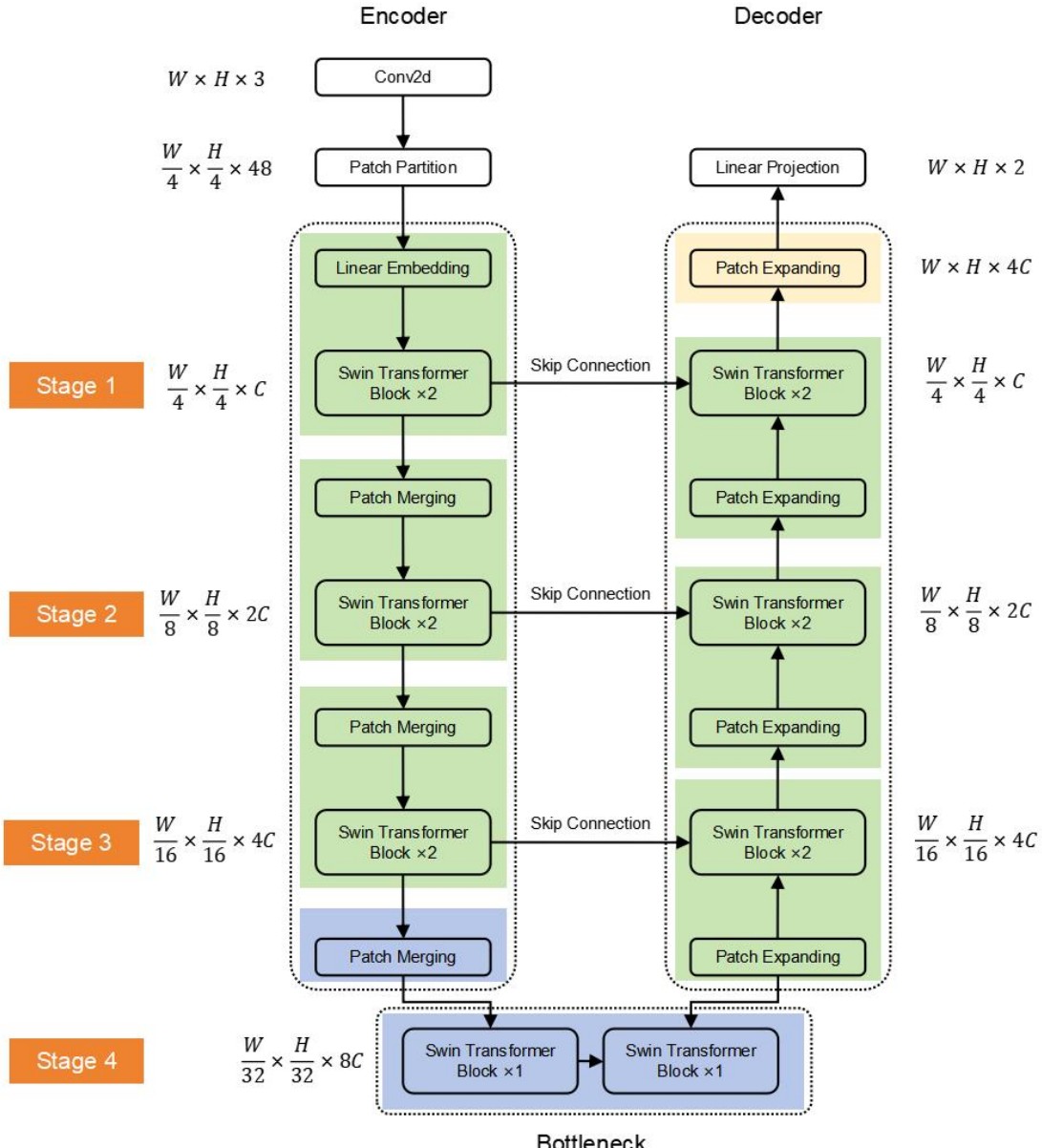

**Figure 3: Structure of SwinUnet Module**

Within the encoder, the patches are inputted into the Swin Transformer block to facilitate learning, while the feature size and resolution stay constant. Simultaneously, the patch merging layer will decrease the quantity of feature maps by a factor of 2 through downsampling, while doubling the feature dimension compared to its original size. This step will be iterated three times in the encoder.

The symmetric decoder, which relies on the Swin Transformer block, serves as the counterpart to the encoder. The deep features that were recovered are enlarged in the decoder using a patch expanding layer. The patch expanding layer transforms the feature maps of adjacent dimensions into higher

resolution feature maps (2× up-sampling) and reduces the feature dimensions by half. In order to prevent the failure of convergence in a deep Swin Transformer block, the bottleneck is constructed using only two SW-MSA modules. This construction ensures that the feature size and resolution stay unchanged. Like UNet, skip connections are employed to integrate multiscale information from the encoder with up-sampled features. Shallow and deep features are linked together to reduce the loss of

spatial information caused by downsampling. Ultimately, the feature map's resolution is increased four times by utilizing the final patch expanding layer, resulting in a restoration to the original input resolution. Afterwards, a linear projection layer is used to generate pixelwise predictions using the upsampled features.

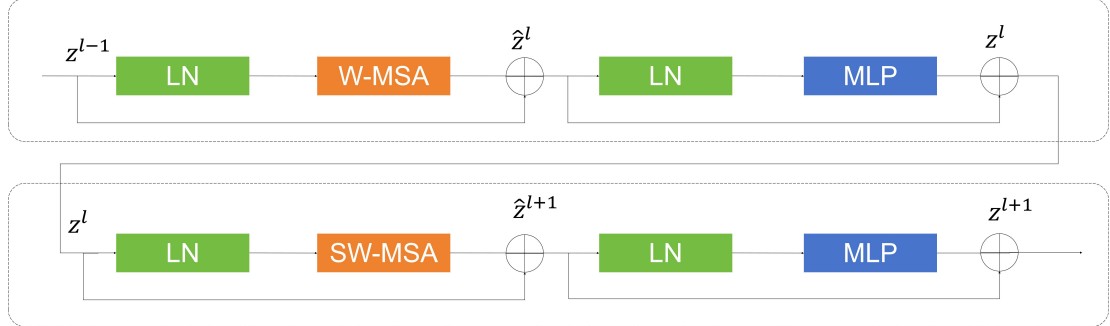

**Figure 4: Structure of Swin Transformer Block**

The fundamental element of SwinUnet is the Swin Transformer block (Fig. 4). The construction

of the Swin Transformer block is based on the concept of the shift window. The Swin Transformer block is composed of two normalization layers (LNs), a multihead self-attention module, residual connections, and a multilayer perceptron (MLP) layer with a GELU nonlinear activation function (Xiao et al., 2020). The use of the window-based multihead self-attention module (W-MSA) and the shift window-based multihead self-attention module (SW-MSA) is observed in two consecutive transformer

blocks (Fig. 4). The formula for the block can be represented as follows (Sheng et al., 2022):

$$\bar{z}^l = W - MSA\left(LN\left(z^{l-1}\right)\right) + z^{l-1} \tag{4}$$

$$z^l = MLP\left(LN\left(\bar{z}^l\right)\right) + \bar{z}^l \qquad (5)$$

$$\bar{z}^{l+1} = SW - MSA\left(LN\left(z^l\right)\right) + z^l \qquad (6)$$

$$z^{l+1} = MLP\left(LN\left(\bar{z}^{l+1}\right)\right) + \bar{z}^{l+1} \qquad (7)$$

The symbol $\bar{z}^l$ denotes the features generated by the (S)W-MSA module, whereas $z^l$ represents the features generated by the MLP module. The variable $l$ corresponds to the number of blocks.

The W-MSA module initially partitions the feature map into several windows based on the specified $M \times M$ dimensions. It subsequently computes the self-attention within each window independently. Nevertheless, the W-MSA module lacks the capability to transfer information between windows. Therefore, it becomes imperative to implement SW-MSA, which relies on shifted windows, in order to address this limitation. The SW-MSA module, together with the W-MSA module in the Swin Transformer block, forms a two-tier structure through which information can be passed through neighboring windows.

The features within each (shift) window are weighted by (S)W-MSA, and the attention weights are adaptively changed via bootstrap feature selection. This approach generates a more extensive feature expression as follows (Zhang et al., 2023):

$$Attention(Q, K, V) = softmax\left(\frac{QK^T}{\sqrt{d}} + B\right)V \qquad (8)$$

where $Q$ denotes the query vector, $K$ denotes the key vector, $V$ denotes the value vector, $d$ denotes the dimensionality of the key–value vector, and $B$ denotes the relative position bias.

The configuration of SwinUnet is shown in Table 1. The downsampling (upsampling) rate refers to the frequency at which upsampling and downsampling are carried out by the patch merging layer and patch expanding layer. After resampling, the output feature maps for each stage have heights and widths of $[560 \times 560, 280 \times 280, 140 \times 140, 70 \times 70]$ accordingly. The window size for performing MSA and SW-MSA operations is set to 7×7. As a result, each stage contains a total of [6400, 1600, 400, 100] windows. The hidden size refers to the length of the vector associated with each token, which represents the feature dimension of the feature map. Upon traversing the linear embedding layer, the feature dimension of the feature map in Unet's stage 1 is augmented to 96, and thereafter doubles in

size in the following stages. The depth refers to the quantity of W-MSA and SW-MSA modules present in the Swin Transformer block. Specifically, in the first three stages, the Swin Transformer block is composed of a double layer structure consisting of one W-MSA module and one SW-MSA module. In stage 4, often known as the bottleneck, there is only one SW-MSA module. The MLP size refers to the number of nodes in the first fully-connected layer of the MLP module, which is four times the hidden size. The "heads" parameter represents the number of nodes in both the W-MSA and SW-MSA in the Swin Transformer block.

Table 1 Detailed architecture configurations of SwinUnet

|  | Downsampling/Upsampling Rate (Output Feature Map Size) | Window size | Window Numbers | Hidden Size | Depth | MLP Size | Heads |
|---|---|---|---|---|---|---|---|
| Stage 1 | 4 (560×560) | 7×7 | 6400 | 96 | 2 | 384 | 3 |
| Stage 2 | 8 (280×280) | 7×7 | 1600 | 192 | 2 | 768 | 6 |
| Stage 3 | 16 (140×140) | 7×7 | 400 | 384 | 2 | 1536 | 12 |
| Stage 4 | 32 (70×70) | 7×7 | 100 | 768 | 1 | 3072 | 24 |

**2.2.4 Statistical tests**

The performance of the model was assessed using various statistical metrics, such as the root mean square error (RMSD), correlation coefficient ($R^2$), and bias. The following formulas were used:

$$RMSD = \sqrt{\frac{1}{N}\sum_{i=1}^{N}(x_i - y_i)^2} \tag{9}$$

$$R^2 = 1 - \frac{\sum_{i=1}^{N}(x_i - y_i)^2}{\sum_{i=1}^{N}(x_i - \overline{x})^2} \tag{10}$$

$$Bias = \frac{1}{N}\sum_{i=1}^{N}(x_i - y_i) \tag{11}$$

where $x_i$ is the true value of pixel $i$, $y_i$ is the predicted value of pixel $i$, and $\overline{x}$ is the average of all true values.

## 3. Results

### 3.1 Comparison of Different Reconstruction Models

The presence of missing values is pervasive in the routine outputs of satellite observations. The occurrence of missing values inside the basin area is primarily impacted by cloud cover. In contrast to that in the sea basin area, the likelihood of missing observations occurring is greater in the nearshore region. In conjunction with cloud cover, anthropogenic activities significantly influence the nearshore region, resulting in elevated chlorophyll-a concentrations and suspended sediment in the water column. Consequently, distinguishing between the intense backscattering caused by algal or nonalgal particulate matter and the scattering of atmospheric aerosols becomes challenging. This difficulty in correcting the atmosphere of nearshore remotely sensed data contributes to the relatively inferior quality of remote sensing in this area. This study aimed to evaluate the performances of several reconstruction schemes by applying them to three satellite observation products characterized by distinct missing data rates. The selected methods for reconstruction include DINEOF, OI, Unet, and OI-SwinUnet. The objective is to assess the effectiveness of these schemes in reconstructing missing data. The imaging of the data took place on three separate occasions: February 11, 2014, February 27, 2015, and January 5, 2016. The spatial coverage of the legitimate data for each respective category is 11.9%, 49.8%, and 66.5%. These percentages indicate the extent to which the observed data are contaminated, ranging from severe to moderate and to least. Within the dataset, it is evident that the data pertaining to severe contamination exhibit a lack of contiguous large valid data in the basin area, as well as a near absence of data in the nearshore area. The dataset exhibits a moderate level of contamination and contains a significant quantity of missing data points within the nearshore region, namely, in the northern and southwestern sectors of the South China Sea. The dataset with the lowest pollution levels has a minimal number of missing data points within the basin area, whereas only a limited number of missing data points are observed in the northern and southeastern regions of the South China Sea (Fig. 5).

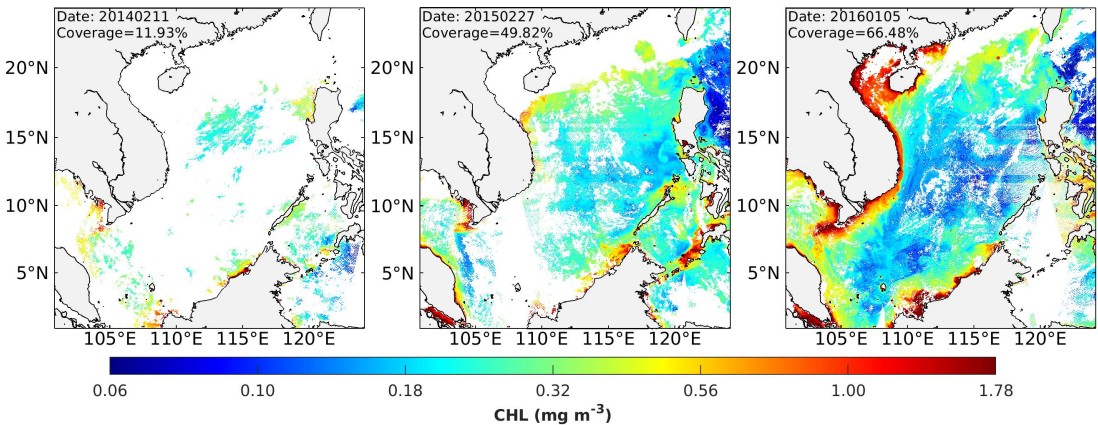

**Figure 5: Satellite observed Chlorophyll-a concentration products with different data coverage**

Figure 6 shows the spatial distributions of the reconstructions obtained from the DINEOF, OI, Unet, and OI-SwinUnet methods for three distinct satellite observation datasets characterized by varying missing data rates. The chlorophyll-a concentration has a coherent structure throughout all the reconstructions, with the exception of Unet. All three reconstructions demonstrate a certain degree of coherence with the original satellite observations, while Unet exhibits a notable tendency to

underestimate the chlorophyll-a concentration in the sea basin region. One possible explanation is that remote sensing images with high spatial resolution (1 km) and extensive coverage encompass a substantial amount of data. However, the Unet model, which relies on convolutional operations, faces limitations due to its own capacity, making it challenging to accurately predict pixel-level images in this context. In relation to the precise replication of visual representations, as evidenced in this research

through the depiction of mesoscale and small-scale phenomena in the marine environment, our proposed OI-SwinUnet exhibits the highest level of efficacy, followed by OI and, ultimately, DINEOF. In general, the outcomes of the DINEOF method exhibit a lower level of clarity than alternative models. This diminished clarity is attributed to the algorithmic characteristics inherent to the DINEOF approach, resulting in the loss of significant mesoscale and small-scale details. The OI-SwinUnet method,

however, is modified by using the OI technique to achieve more plausible reconstruction outcomes.

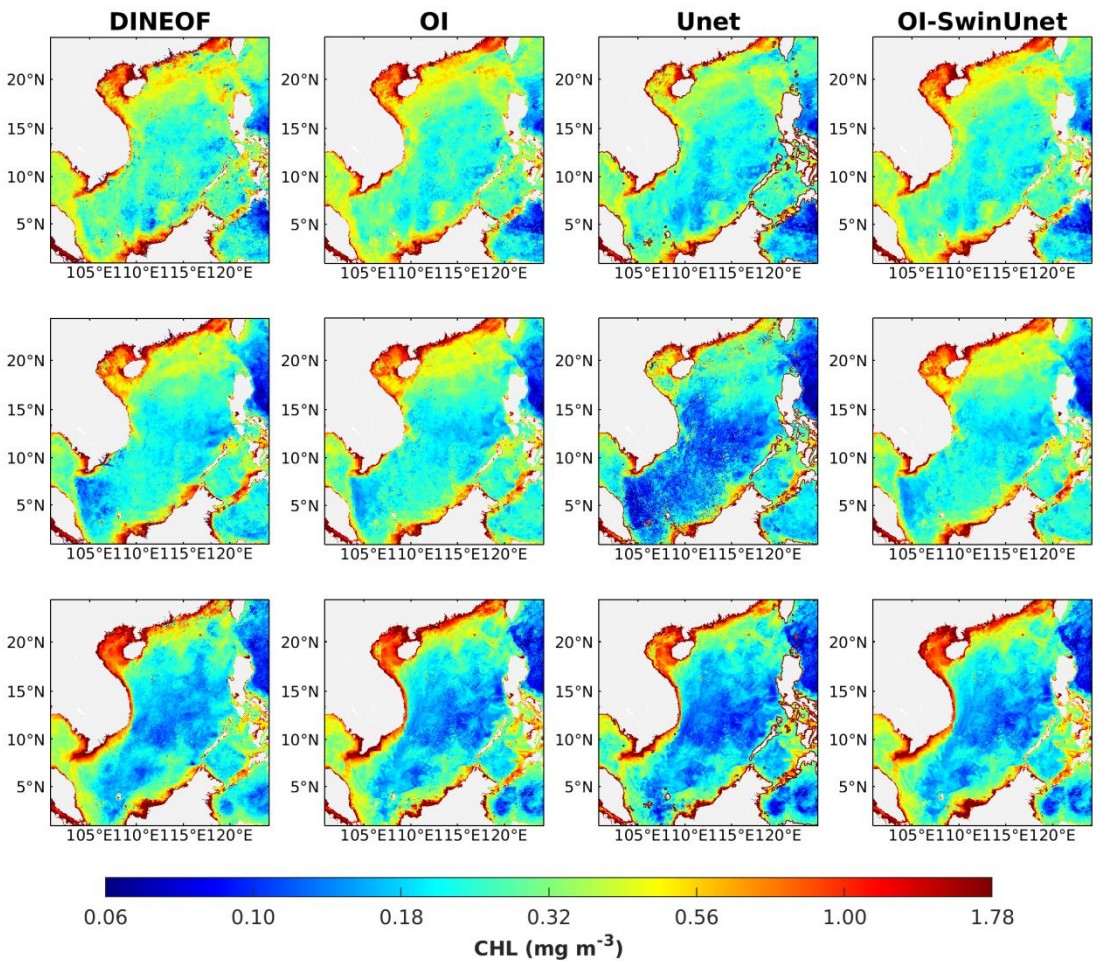

**Figure 6: The reconstructed CHL based on different models, data acquired on (Upper panel: 11th February, 2014. Middle panel: February 27, 2015. Lower panel: January 5, 2016)**

The scatterplot confirms the earlier observation that our suggested OI-SwinUnet model outperforms the other three reconstruction models in terms of different performance metrics, including RMSD, $R^2$, and bias. This holds true regardless of the chlorophyll-a concentration product utilized for coverage (Fig. 7). The slope of the trend line for the Unet results indicates that the Unet reconstruction method is likely to significantly underestimate low chlorophyll values and overestimate high chlorophyll values. This discovery aligns with the prior finding that the Unet reconstruction outcomes for the Sea Basin region are notably inferior to those of the other methods.

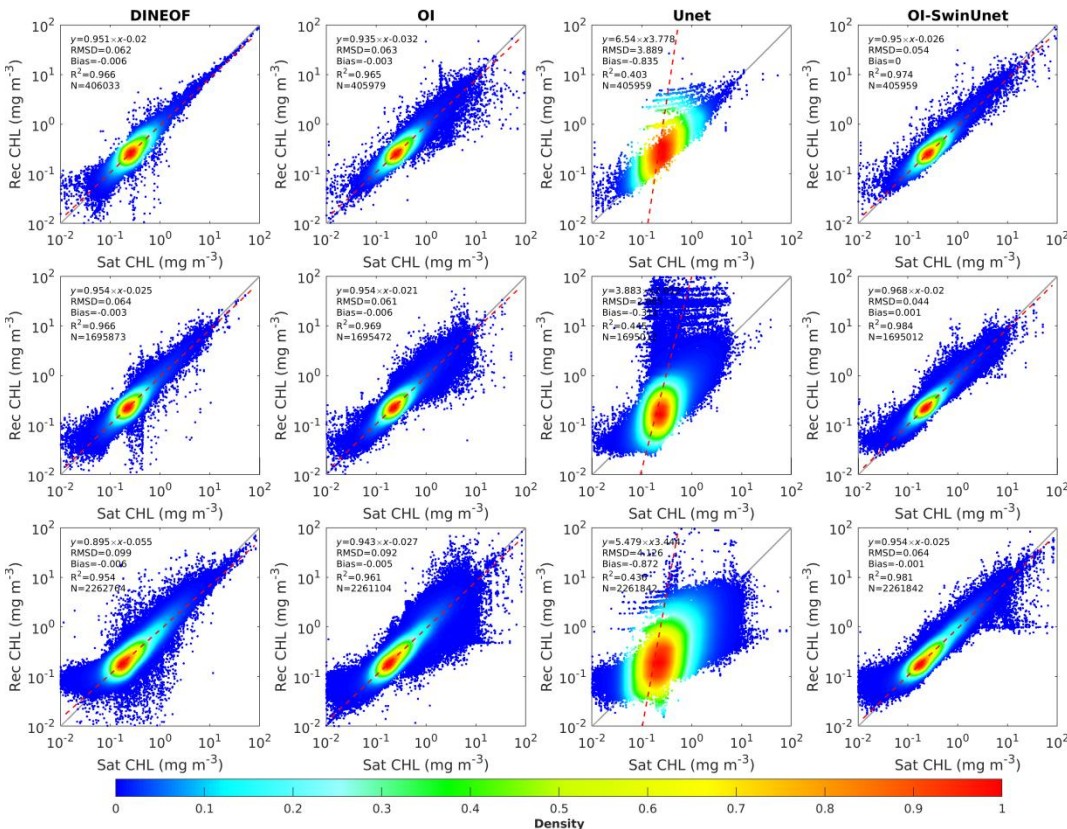

**Figure 7: Scatter plots between satellite-derived (merged products from Aqua and Terra) and reconstructed chlorophyll-a concentration of different models, data acquired on (Upper panel: 11th February, 2014. Middle panel: February 27, 2015. Lower panel: January 5, 2016)**

To better understand the effectiveness of the various reconstruction methods, we conducted a statistical analysis on the observed and reconstructed complete dataset, which included daily products from 2013 to 2017. The average RMSD values derived from the four methods (DINEOF, OI, Unet, and OI-SwinUnet) were 0.09 mg m⁻³, 0.06 mg m⁻³, 0.13 mg m⁻³, and 0.06 mg m⁻³, respectively. The OI-SwinUnet and OI algorithms exhibited the most superior performance, with DINEOF ranking second, while Unet yielded the least favorable outcomes. The upper and lower quartiles Q3 and Q1 had ranges of 0.08-0.12 mg m⁻³, 0.04-0.10 mg m⁻³, 0.12-0.16 mg m⁻³, and 0.05-0.07 mg m⁻³, respectively. The box plots show that the OI-SwinUnet reconstruction scheme consistently produces reasonable reconstruction results across all the datasets. The average $R^2$ values for the four systems are 0.90, 0.96, 0.85, and 0.96. The bias is nearly zero for all three methods, except for Unet. The box plots for $R^2$ and bias exhibit comparable features to the box plots for RMSD (Fig. 8). The findings demonstrate that OI-SwinUnet outperforms the other methods in reconstructing daily satellite observation products for

the period from 2013 to 2017.

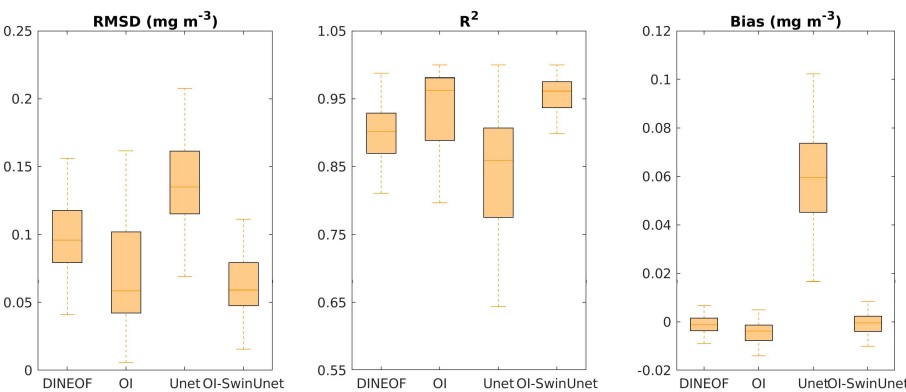

**Figure 8: Boxplots of model performance**

Two representative pixels were sampled from the Pearl River Estuary in the northern part of the South China Sea and the central part of the South China Sea. These pixels were chosen to represent highly turbid water and clean water, respectively (red triangles in Fig. 10). The purpose was to compare the performance of OI-SwinUnet and three other methods in terms of filling gaps in time series data.

The results indicate that our proposed OI-SwinUnet demonstrates strong resilience to localized extremes, typically outliers. Within the clean water region, the OI-SwinUnet, DINEOF, and OI methods are capable of analyzing the dynamic patterns of the chlorophyll time series. However, the Unet method performs slightly less accurately, as it tends to underestimate chlorophyll values in most time intervals. This discrepancy is particularly evident in time intervals where satellite observations are

consistently absent. In areas with high levels of turbid water, the OI-SwinUnet method performs similarly to the DINEOF method during periods of consecutive high chlorophyll levels. Figure 9 demonstrates that DINEOF is more successful in reconstructing high chlorophyll levels. This suggests that the method can effectively fill in the gaps in the time series data, allowing for reasonable patterns of interannual variation in chlorophyll-a to be observed.

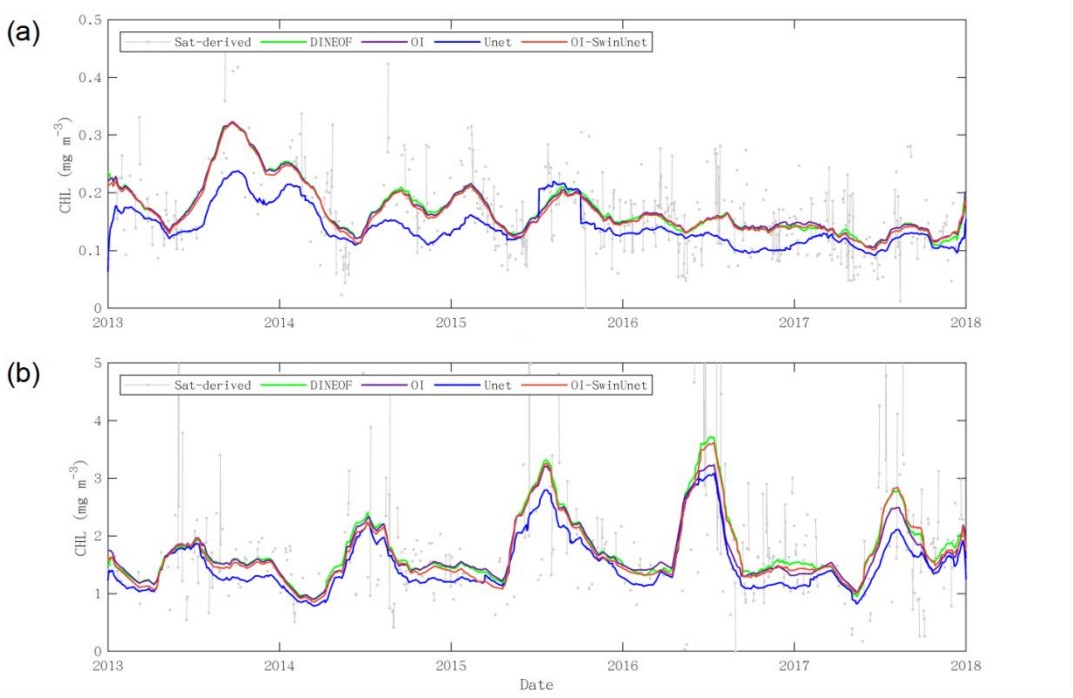

**Figure 9: Gap-filled time series of two represented pixels (a) clear water, and (b) high turbid water using DINEOF , OI, Unet and OI-SwinUnet methods**

375  The performance of OI-SwinUnet was assessed by partitioning the study area into three zones according to water depth: nearshore, shelf, and basin. The water depths in the nearshore zone varied between 0 and -50 m, while the water depths in the shelf zone ranged from -50 m to -1000 m (Fig. 10). In the basin zone, the water depths were less than -1000 m. The allocation of the maritime territory included the influence of local features, ocean currents, and the distribution of biological ecosystems.

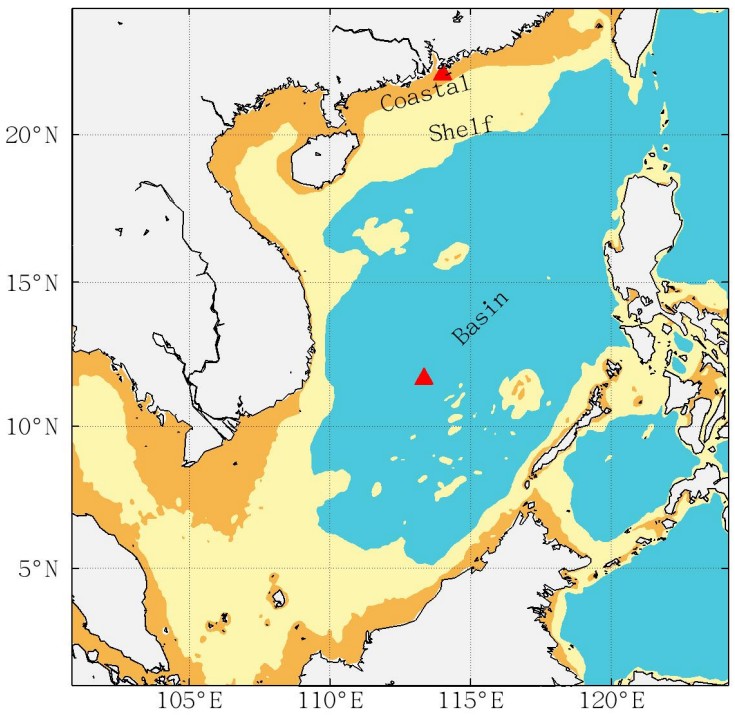

**Figure 10: Three divided regions in South China Sea**

The nearshore zone had a median RMSD value of 0.075 mg m$^{-3}$, the shelf zone had a median

RMSD value of 0.065 mg m$^{-3}$, and the basin zone had a median RMSD value of 0.07 mg m$^{-3}$. The R$^2$

values were 0.94, 0.96, and 0.95 in the nearshore, shelf, and basin zones, respectively. In the nearshore

zone, the median value of bias was $2\times10^{-3}$ mg m$^{-3}$, while it was approximately 0 in both the shelf zone

and the basin zone (Fig. 11). OI-SwinUnet exhibited superior performance in the shelf zone, followed

by the basin zone, and demonstrated subpar performance in the nearshore zone. The model's inferior

performance in the nearshore region can be attributed to the limited amount of training data available.

The nearshore zone has the highest proportion of missing data compared to the other zones, with a

median percentage of valid data of approximately 21%. In contrast, the median values for the other

zones ranged from 25% to 30%. A small amount of training data hinders the model's ability to

understand the characteristics of the location.

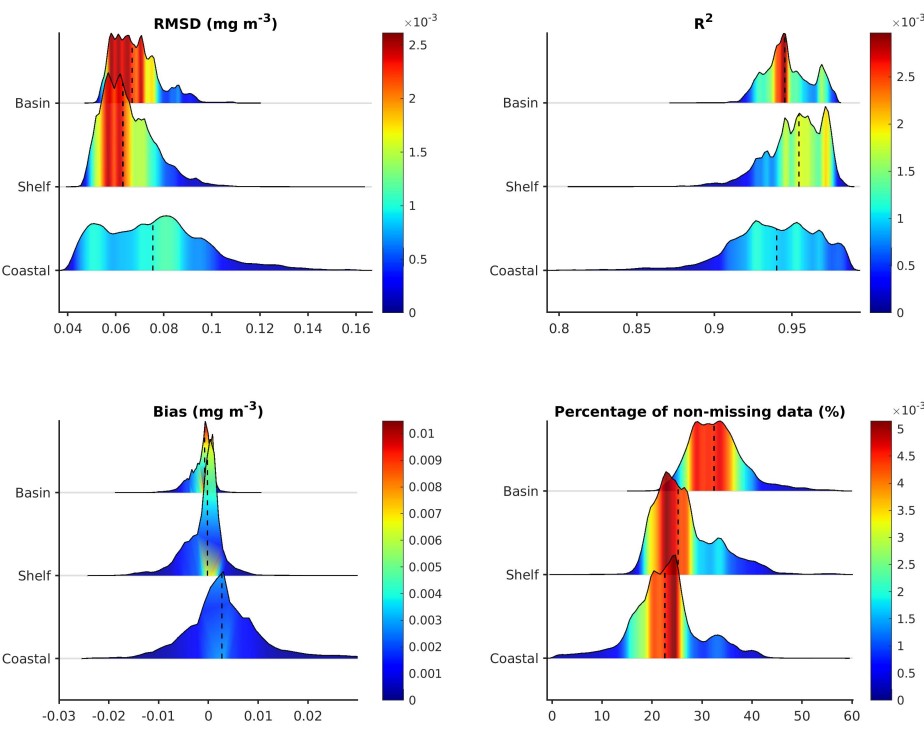

**Figure 11: Performance of OI-SwinUnet in different regions**

## 3.2 Model Robustness

Missing patterns in remotely sensed data are a crucial consideration when evaluating the generalizability of reconstructed models (Bessenbacher and Seneviratne and Gudmundsson, 2022). In the context of data analysis, the phenomenon of missing data can be classified into three main categories (Rubin, 1976): missing completely at random (MCAR), missing at random (MAR), and missing not at random (MNAR). The next sections provide a description of the aforementioned categories that are absent, with specific reference to satellite observations.

1) The phenomenon of missing data is classified as completely random when the likelihood of missing data points is independent of any underlying mechanism (MCAR, Fig. 12a). Random sensor failures can contribute to missing data in satellite observations, although they are typically not the primary cause of missing data.

2) Missing data can frequently occur in satellite scans due to the absence of satellite orbits over specific places during particular periods. The likelihood of encountering missing data points is not influenced by the specific value of the missing data point. The aforementioned pattern is referred

to as "missing at random" (MAR), as depicted in Fig. 12b.

3)    One of the most intricate patterns of missingness is known as missing not at random (MNAR). The masking of data points in this scenario is contingent upon the presence of missing data. The observed variables might influence this mechanism, such as when the values surpass or fall below a specific threshold, rendering them unobservable (see Fig. 12c).. When utilizing satellite

observations to measure sea surface chlorophyll-a concentrations in the visible band, it is important to note that the reflectance of clouds does not represent signals originating from the water column at the sea surface. Consequently, during the preprocessing of data, pixels that are covered by clouds are identified and excluded from further analysis. In this context, it is not justifiable to assume that the chlorophyll-a concentration under cloudy conditions is not

significantly different from the observed chlorophyll-a concentration. Consequently, it is not appropriate to assume statistical independence between the missing data points and the unobserved values associated with those missing points.

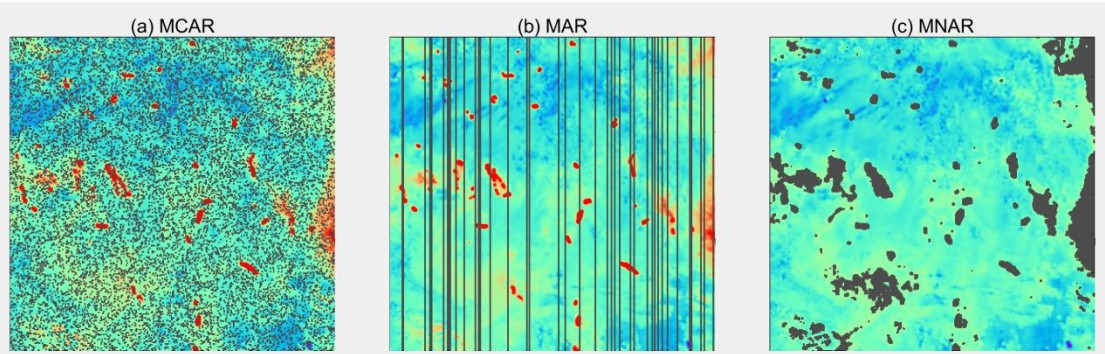

**Figure 12: Illustration of three missing patterns: (a) MCAR, (b) MAR, (c) MNAR**

The presence of missing values in satellite observations is a prevalent issue since these data often constitute a substantial portion of the data. These missing values exhibit a complex pattern known as

MNAR, which poses challenges for gap filling methods. Hence, our research aimed to investigate the extent to which an increase in data sparsity and an increase in complexity of missing cases result in a decrease in the efficacy of OI-SwinUnet.

For our trials, we selected the OI-SwinUnet reconstructed data from April 26, 2014. The data exhibit a genuine missing data rate of 48%, with a predominant concentration of missing pixels in the

northern and southwestern regions of the South China Sea. The reconstructed data with varying missing data rates were generated by initially removing a portion of the whole reconstructed dataset

based on three different patterns. Subsequently, the new data were reintroduced into the trained network. The missing data were defined within a range of 10% to 90%, with increments of 10%. The performance metrics utilized in the experiment included the RMSD, $R^2$, and bias. These metrics were employed to compare the observed and reconstructed values.

This case study revealed that the disparities between reconstructed and satellite observations of chlorophyll-a concentrations in the basin area exhibit a consistent level of stability across various rates of missing data. The discrepancy between the two measurements reached approximately $\pm 2 \times 10^{-3}$ mg m$^{-3}$. In instances where the rate of missing data is elevated, there is a notable disparity between the reconstructed and observed values, particularly within the nearshore sea region characterized by high chlorophyll-a concentrations. Areas exhibiting differences exceeding 0.01 mg m$^{-3}$ are predominantly situated in the nearshore region of the Philippines, specifically in the eastern sector of the South China Sea. As the incidence of missing data decreases, the quality of the reconstructed results constantly improves. Consequently, the areas in the nearshore region of the eastern South China Sea that previously exhibited significant differences nearly vanished (Fig. 13).

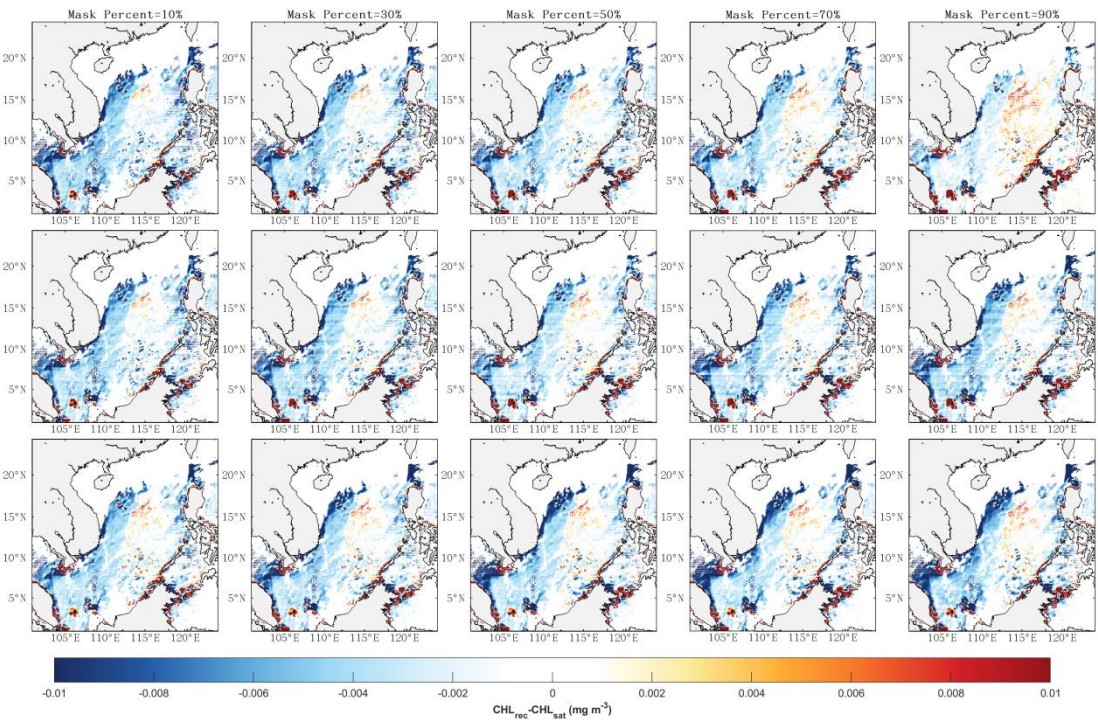

**Figure 13: Spatial distribution of the OI-SwinUnet reconstruction in three different patterns for missing**

**data rates ranging from 10% to 90% (upper panel: MCAR; middle panel: MAR; lower panel: MNAR)**

As the percentage of deletions increased from 10% to 90%, the RMSD of the MCAR system increased from 0.049 to 0.055 mg m$^{-3}$. Additionally, R$^2$ fell from 0.974 to 0.967, and the bias exhibited variation ranging from -1.6×10$^{-3}$ to 4.9×10$^{-3}$ mg m$^{-3}$. The RMSD of the MAR pattern increased from 0.049 to 0.056 mg m$^{-3}$. Additionally, R$^2$ decreased from 0.974 to 0.967. Furthermore, the range of the

bias spanned from 1.5×10$^{-3}$ to 3.9×10$^{-3}$ mg m$^{-3}$. The RMSD of the MNAR pattern increased from 0.052 to 0.055 mg m$^{-3}$. Additionally, R$^2$ fell from 0.968 to 0.965. The bias varied from 2.5×10$^{-3}$ to 3.5×10$^{-3}$ mg m$^{-3}$. The performance of the model in the real situation was superior to that of the artificial missing patterns, both operating at the same missing data rate (represented by dots in Fig. 14). In contrast to the other two patterns, the MCAR pattern more easily estimates missing values due to the presence of

similar observations neighboring the missing values. The MAR pattern exposes a large missing patch that is inadequately addressed by spatiotemporal interpolation, resulting in a decrease in gap-filling performance when compared to that of MCAR. The MNAR pattern represents the most intricate missing pattern, thus capturing the upper limit of performance achievable in real-world scenarios for OI-SwinUnet.

In general, the slight disparity observed between the metrics obtained from the simulated missing mode and the actual scenario serves to illustrate the maximum operational performance of OI-SwinUnet when real data are used. The excellent performance of the model may be attributed to the inclusion of input data, which included both temporal and spatial dimensions. Specifically, the input data incorporate information pertaining to the previous 15-day period as well as the post 15-day period.

Despite the intentional masking of the original satellite observation data on the same day, OI-SwinUnet is capable of acquiring knowledge about various spatial scales by utilizing information from both before and after the same day. Consequently, the proposed method can successfully reconstruct missing regions with a high level of accuracy.

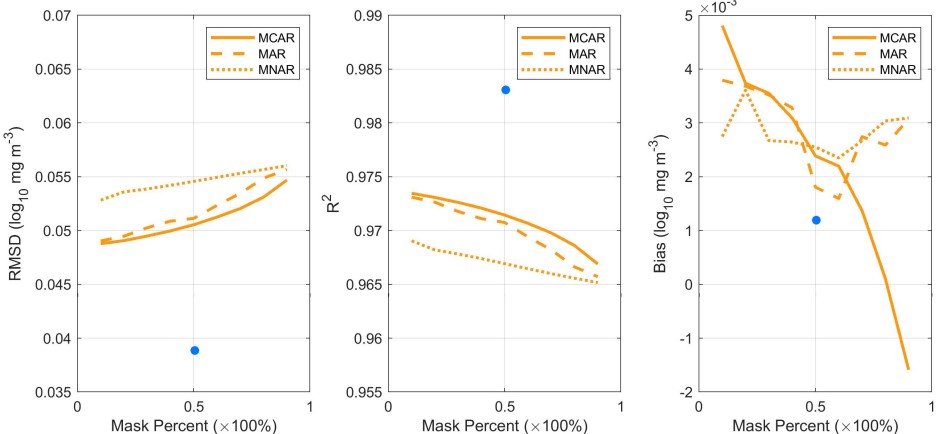

**Figure 14: The OI-SwinUnet reconstruction performance in three different patterns for missing data rates ranging from 10% to 90%**

## 4. Discussion

### 4.1 Spatiotemporal Characteristics

For a gap-filling framework to possess utility in scientific and practical contexts, it is imperative that this framework possesses the capacity to accurately recapitulate the essential characteristics of the phenomena under investigation. Oceans exhibit a multitude of mesoscale and small-scale phenomena, including eddies, upwelling, fronts, and other similar processes. Oceanic dynamic mechanisms play a crucial role in regulating the proliferation and decline of phytoplankton populations within the marine environment. On the other hand, the examination of phytoplankton dispersion on the ocean surface using satellite observations can enhance the understanding of intricate and localized dynamic processes. This study investigated the potential benefits of utilizing reconstructed datasets obtained through OI-SwinUnet for monitoring seasonal ocean phenomena.

The spatial distribution of the seasonal mean chlorophyll-a concentration, as depicted in Fig. 15, exhibited a consistent pattern over the four seasons. Notably, higher values were observed in nearshore waters than in offshore waters. This phenomenon could be attributed to the variability in nutrient content originating from terrestrial sources. Phytoplankton in nearshore areas typically consume nutrients, resulting in decreased nutrient concentrations in offshore seas upon their arrival (Bristow et al., 2017; Liu et al., 2003). During the summer season, a region of significant ecological importance was observed along the coastal area of Vietnam due to the occurrence of upwelling. This zone extends

from the southeastern part of Vietnam toward the east. In this region, the average concentration of chlorophyll-a is approximately 0.30 mg m$^{-3}$. In contrast, the surrounding basin area exhibited an average chlorophyll-a concentration of approximately 0.16 mg m$^{-3}$ during the same time period. The

upwelling along the eastern coast of Vietnam forms during May-September and reaches its mature stage during July-August (Fang et al., 2012; Voss et al., 2006). Past studies have shown that this summer upwelling is always in the form of a jet-like cold tongue (or cold patch) originating off the coast of Vietnam between 9° and 15° N (Hein et al., 2013). After generation, the cold tongue may extend eastward or northeastward into the central South China Sea (Gan et al., 2006). In the offshore

region, the upwelling is usually accompanied by the Vietnamese cold eddy (Hu and Wang, 2016). In addition to the cold water observed in upwelling regions, high chlorophyll-a concentrations are often reported (Ho et al., 2000; Li et al., 2014; Zhao and Tang, 2007). By providing deep nutrient-rich water, upwelling stimulates the growth of phytoplankton in the euphotic zone, thus significantly altering the trophic state of the Vietnamese nearshore region (Bombar et al., 2010). Driven by transport in offshore

currents, upwelling nutrients and stimulated high Chl-a ($\geq$0.2 mg m$^{-3}$) can extend from the coast to 116°E, creating 'Chl-a jets' (Chen and Xiu and Chai, 2014).

In the winter, the average concentration of chlorophyll-a in the northern region of the South China Sea was greater than that in the southern region of the South China Sea. The average concentration of chlorophyll-a in the northern region of the South China Sea is approximately 0.32 mg m$^{-3}$. This

concentration is particularly prominent in the northwestern area of the Luzon Strait, where the average chlorophyll-a concentration can reach as high as 0.50 mg m$^{-3}$. Previous research has indicated a correlation between the occurrence of phytoplankton blooms in the northern region of the South China Sea during winter and the deepening of the mixing layer caused by intensified winds and cooling of the sea surface (Shen et al., 2008; Liu and Chen, 2014). The high concentration of nutrients brought by

deep water encourages the growth of surface phytoplankton.

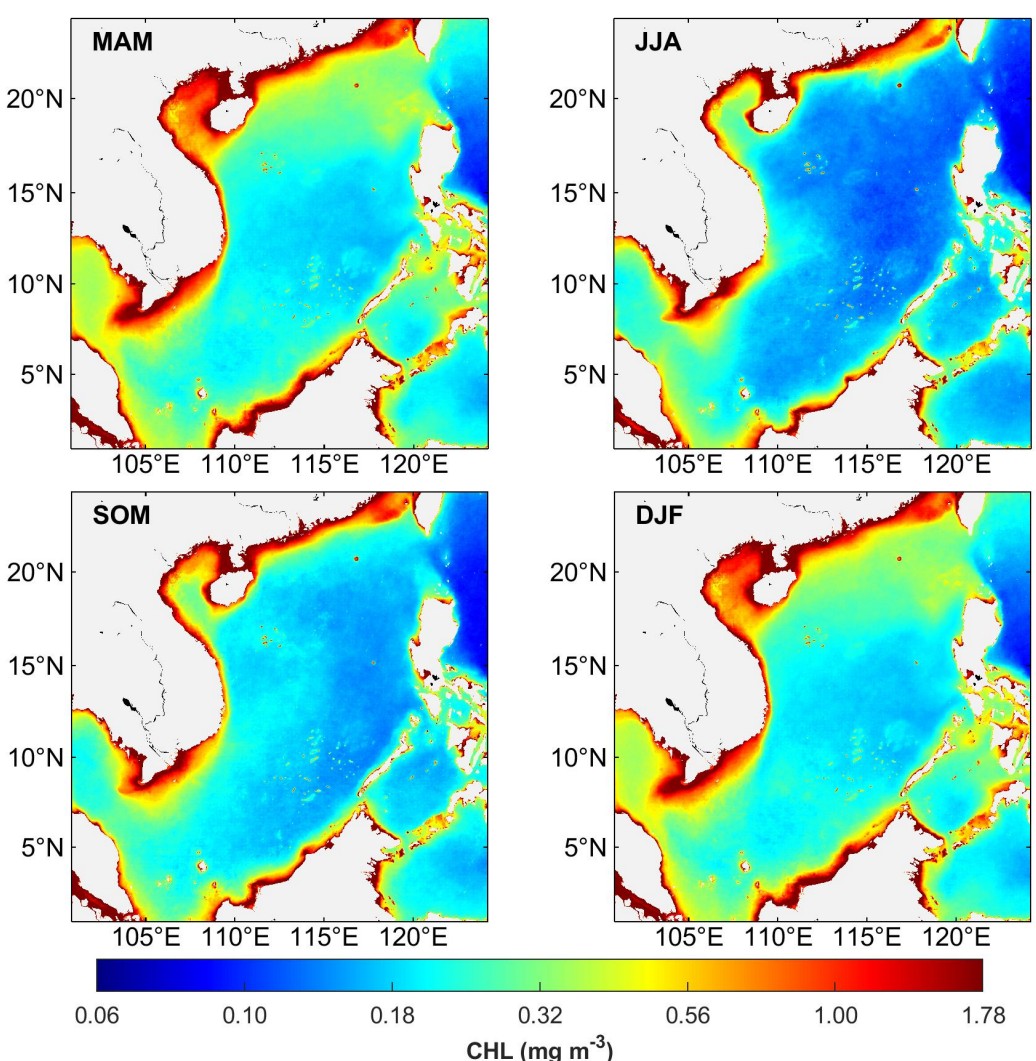

**Figure 15: Spatial distribution of reconstructed chlorophyll-a concentration in SCS during four seasons**

Figure 16 illustrates the fluctuations in chlorophyll-a concentrations in various parts of the South China Sea across the four seasons. Typically, the South China Sea experiences elevated surface chlorophyll-a concentrations during winter and reduced levels throughout spring or summer. The average chlorophyll-a content in the South China Sea ranged from 0.35 to 0.46 mg m$^{-3}$. The nearshore area exhibited the greatest average chlorophyll-a concentration, ranging from 0.93 to 1.21 mg m$^{-3}$. This is attributed to the significant impact of human activities in this particular area. The basin zone exhibited the smallest range of mean chlorophyll-a concentrations, which varied from 0.14 to 0.22 mg m$^{-3}$. Additionally, there was minimal seasonal fluctuation in the depth of the mixed layer in this area. Throughout all four seasons, the subsurface layer consistently contained zones with high chlorophyll-a concentrations (Zhang et al., 2016). In general, the reconstructed dataset successfully replicated the

seasonal-scale geographical and temporal patterns of chlorophyll-a concentrations in the surface layer
of the South China Sea.

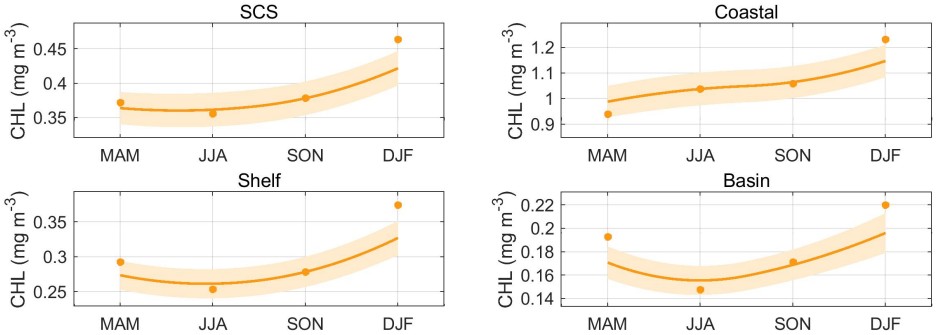

**Figure 16: Seasonal variations of reconstructed chlorophyll-a concentration in SCS and the divided seven regions**

## 4.2 Responses to Small-scale and mesoscale dynamical processes

In addition to the aforementioned seasonal-scale processes, a number of recent studies have
provided evidence for the occurrence of weather-scale phytoplankton outbreaks in the South China Sea.
To comprehensively examine the seasonal variations in chlorophyll-a concentrations at the surface of
the South China Sea over an extended period, it is necessary to utilize extensive time series data.
Conversely, the detection of phytoplankton outbreaks occurring at shorter timescales, such as
weather-scale events, is relatively straightforward due to their prominent signals. Consequently,
numerous studies have concentrated on investigating these sporadic or localized phytoplankton
outbreaks. Nevertheless, obtaining weather-scale information that encompasses the entire research area
through remotely sensed observations might be challenging. The attainment of comprehensive
knowledge and comprehension pertaining to the ocean is evidently challenging when exclusively
relying on satellite observations that exhibit a substantial quantity of data gaps. Through the utilization
of the OI-SwinUnet reconstructed dataset for chlorophyll-a concentration, we were able to visually
perceive and examine localized episodic occurrences of phytoplankton abnormalities.

This phenomenon can be observed in the manner in which the plume front impacts upwelling
regions (Fig. 17). The reconstructed dataset spans from June 14, 2015, to June 25, 2015, encompassing
the geographical region extending from the Pearl River estuary to southeastern Taiwan. Over the course
of a 12-day period, the initial three days were characterized by the preliminary stage of the frontal

impact on the upwelling phenomenon. During this phase, the plume front began to develop in the eastern direction. The images also reveal the existence of an upwelling zone in the shallow waters of southwestern Taiwan, which can be attributed to topographical alteration. The mean concentration of chlorophyll-a in this region characterized by upwelling is approximately 0.63 mg m$^{-3}$. During days 4 to 6, the second phase of frontal-affected upwelling occurred and was characterized by the arrival of freshwater to the upwelling region. This influx of freshwater leads to a notable increase in the concentration of chlorophyll-a. Specifically, the average chlorophyll-a concentration reaches its peak value of 1.15 mg m$^{-3}$ on day 6. After the seventh day, the subsequent phase, denoted the third stage, ensues. During this period, the freshwater progressively recedes toward the western region, causing its impact zone to gradually withdraw from the upwelling area. However, it is noteworthy that the chlorophyll-a concentration inside the upwelling area remains elevated, exhibiting an average value of approximately 1.00 mg m$^{-3}$. In contrast to that in the initial stage, the mean chlorophyll-a concentration notably increased by approximately 0.37 mg m$^{-3}$ during the subsequent stage, constituting almost 50% of the overall increase. The presence of a substantial quantity of phytoplankton in freshwater environments led to a notable increase in chlorophyll-a levels on the surface of the sea within the upwelling zone. The observed increase in the chlorophyll-a concentration in the upwelling zone continued even after the cessation of freshwater input, indicating that the presence of freshwater can stimulate subsequent phytoplankton proliferation by delivering substantial quantities of nutrients to the upwelling zone.

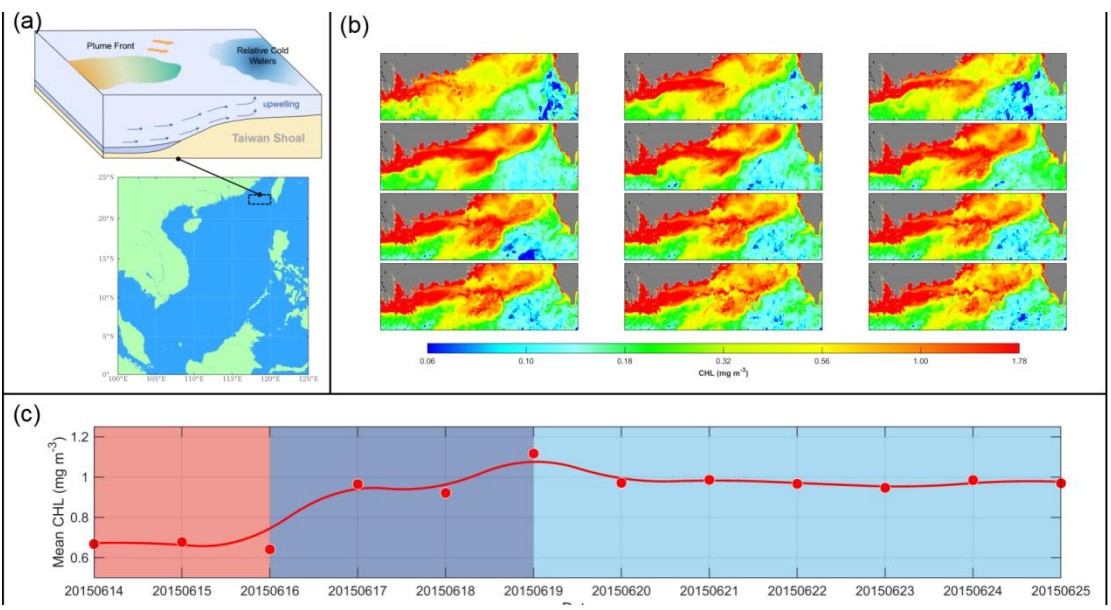

**Figure 17: Pearl River plume front affects surface phytoplankton in the upwelling zone of southwestern Taiwan (a) Location of upwelling; (b) Processes of freshwater affecting the upwelling zone captured by reconstructed daily data; (c) Time series of daily mean chlorophyll-a concentration in the upwelling zone**

Another instance involves the assessment of mesoscale eddies on an individual basis. The interaction between mesoscale eddies and sea surface chlorophyll is intricate and might involve the simultaneous occurrence of several physical phenomena. The frequent and substantial lack of surface chlorophyll data in satellite observations poses challenges in observing and analyzing the ecological impacts of individuals or a small number of eddies. Consequently, numerous studies have concentrated

on examining the comprehensive ecological impacts of mesoscale eddy activity in a specific maritime region and its primary dynamic mechanisms. To assess the effectiveness of the reconstructed chlorophyll product in monitoring the changes in chlorophyll levels within a single mesoscale eddy, we randomly chose an anticyclonic eddy and analyzed it using both the incomplete satellite-observed chlorophyll product and the reconstructed chlorophyll product.

The mesoscale eddy with the identification number #27708868 is an anticyclonic eddy (AE) situated in the offshore region of southeastern Vietnam. The AE originated on April 9, 2016, and ceased to exist on August 19, 2016. The life cycle of AE is 133 days, which can be divided into four distinct stages: generation, strengthening, maturity, and extinction. The AE originated in the western region of the South China Sea at 112.8°E, 13.2°N. Thereafter, the typhoon followed a path toward the north, then

toward the west and then south until it eventually dissipated near 110.2°E, 14.0°N (Fig. 18a). The magnitude of change varied between 5.0 cm and 10.0 cm during the phase of increased intensity. In the mature stage, the amplitude exceeded 15.0 cm, reaching a maximum value of approximately 25.0 cm. During the extinction stage, the amplitude substantially decreased, dropping to less than 5 cm during a span of 25 days. Additionally, the shape and structure of the AE became unclear and difficult to discern

(Fig. 18b).

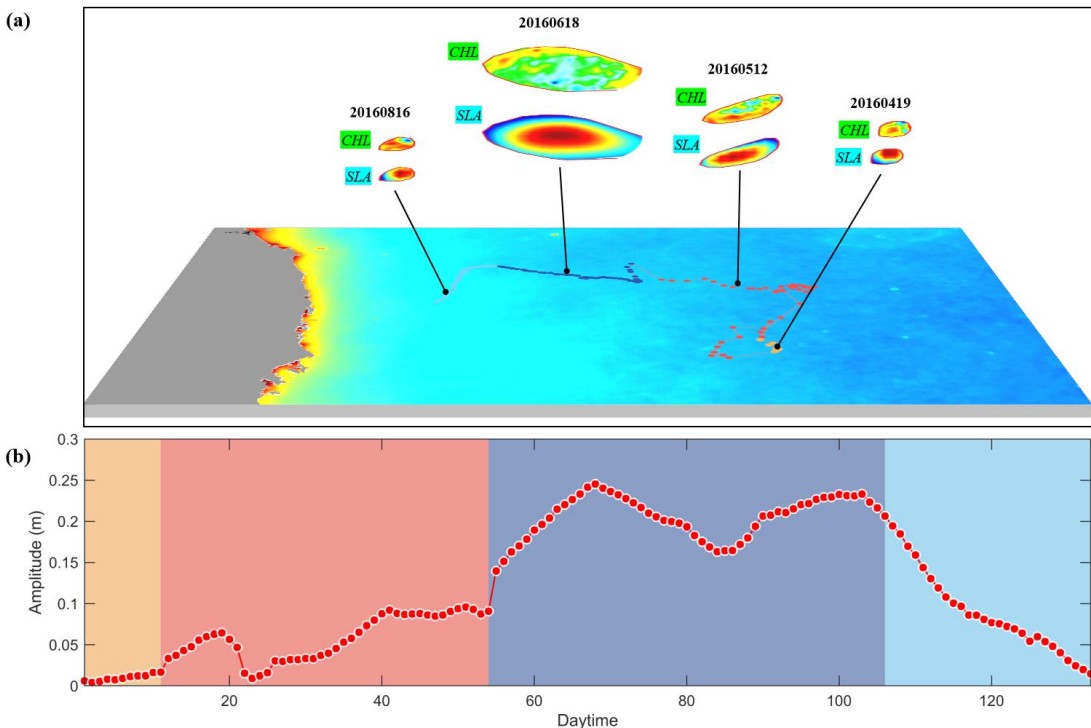

**Figure 18: (a) Trajectory of anticyclonic eddy, sea surface chlorophyll and sea level anomalies within the eddy at different life stages from April to August 2016 in the southeastern Vietnamese Sea (the map is filled in with the mean sea surface chlorophyll-a concentration throughout the life cycle of the eddies); (b) Amplitude plotted against time during the lifespan of the eddy**

The impacts of mesoscale eddies on marine ecosystems can be categorized as eddy-induced pumping (Mcgillicuddy et al., 1998; Siegel et al., 2008), eddy-induced Ekman pumping (Martin and Richards, 2001), or eddy stirring (Chelton et al., 2011). To gather data on the changes in chlorophyll-a concentration in mesoscale eddies, we systematically determined the geographic location of the AEs during each sampling period. We then collected data from satellite observations and reconstructions taken at the same time. Chlorophyll disturbances within a range of ±1.5 times the effective radius of the eddy were interpolated in a linear manner onto a grid that had been normalized. By transforming mesoscale chlorophyll perturbation from a Cartesian coordinate system $CHL_{(x,y)}$ to a polar coordinate system $CHL_{(r,\theta)}$, it is possible to split overall chlorophyll perturbation $CHL_{(r,\theta)}$ into two distinct types: a symmetric dipole ($\overline{CHL_{(r)}}$) and a monopole chlorophyll perturbation ($CHL'_{(r,\theta)}$). The dipole potential is determined by calculating the average chlorophyll-a concentration in the radial direction $\overline{CHL_{(r)}}$. The monopole density was determined by subtracting the dipole energy from the total chlorophyll perturbation.

$$CHL'_{(r,\theta)} = CHL_{(r,\theta)} - \overline{CHL_{(r)}}$$ (12)

where r represents the radius of the eddy and θ represents the angle of direction. $CHL'_{(r,\theta)}$ refers to the dipole chlorophyll perturbation, which can be considered a consequence of the eddy stirring caused by the eddy. $CHL_{(r,\theta)}$ represents the overall change in chlorophyll concentration within the eddy. $\overline{CHL_{(r)}}$ refers to the specific change in chlorophyll concentration caused by processes such as eddy-induced pumping.

Figure 19 displays the outcomes of the perturbations in total chlorophyll and the perturbations in separation within the AE, as computed using both satellite observation data and reconstructed data. Both the reconstructed and satellite observation data indicate a distribution of total chlorophyll perturbations, with higher values in the west and lower values in the east. This distribution aligns with the characteristic westward gradient of background chlorophyll in the western part of the South China Sea, as shown in Fig. 18a. The mean chlorophyll-a concentration within a distance equal to the radius of the eddy's inner core is 0.12 mg m$^{-3}$, as determined from reconstructed data. This value is in close proximity to the mean chlorophyll-a concentration, which is 0.11 mg m$^{-3}$, as computed from satellite observations. The mean chlorophyll-a concentration within a distance of 1-1.5 times the radius of the eddy's outer edge, as determined from reconstructed data, is 0.14 mg m$^{-3}$, which is equivalent to the concentration obtained from satellite data. The analysis of the isolated monopole yielded an average chlorophyll perturbation value of 0.12 mg m$^{-3}$ in the inner core and 0.14 mg m$^{-3}$ in the outer edge, as determined from the reconstructed data. The mean chlorophyll disturbance derived from satellite observations is 0.13 mg m$^{-3}$ in the central region and 0.16 mg m$^{-3}$ in the peripheral area. The reconstructed data exhibited decreases of approximately 7.7% and 12.5% in the mean values within the inner core and outside edge, respectively. The analysis of the separated dipole results reveals that both datasets exhibit positive anomalies in the western region of the eddy and negative anomalies in the eastern region. However, a notable distinction is that the contrast between the eastern and western sides of the eddy is more pronounced in the reconstructed data than in the satellite observation. Additionally, the computed east−west disparity derived from the reconstructed data is greater than that observed from the satellite data.

An issue with utilizing satellite observations to quantify chlorophyll perturbations is the formation of isolated monopole perturbations due to inadequate data. These perturbations are created by the

overlap of many circles with different values. The current situation contradicts the previous assumption that changes in monopole chlorophyll content are caused by vertical transport, such as eddy-induced pumping. Consequently, the chlorophyll-a concentration consistently increases as the radius of the eddy

increases, and the lowest values are found at the center of the eddy. From this perspective, the utilization of reconstructed data to isolate monopole disturbances appears to provide a more comprehensive explanation for the vertical transport characteristics of eddies. However, it is important to note that the average value of monopole chlorophyll disturbances obtained from incomplete satellite data is typically greater than that from reconstructed data. This disparity between the two datasets

cannot be disregarded. A discrepancy in monopole disturbances results in a more significant difference in dipole disturbances, even if the total chlorophyll disturbances calculated from the two datasets do not differ significantly. Hence, the accuracy of the data might directly impact researchers' computations of the proportional influences of horizontal stirring and eddy-induced pumping and, consequently, their assessment of the principal mechanism through which eddies influence sea surface chlorophyll.

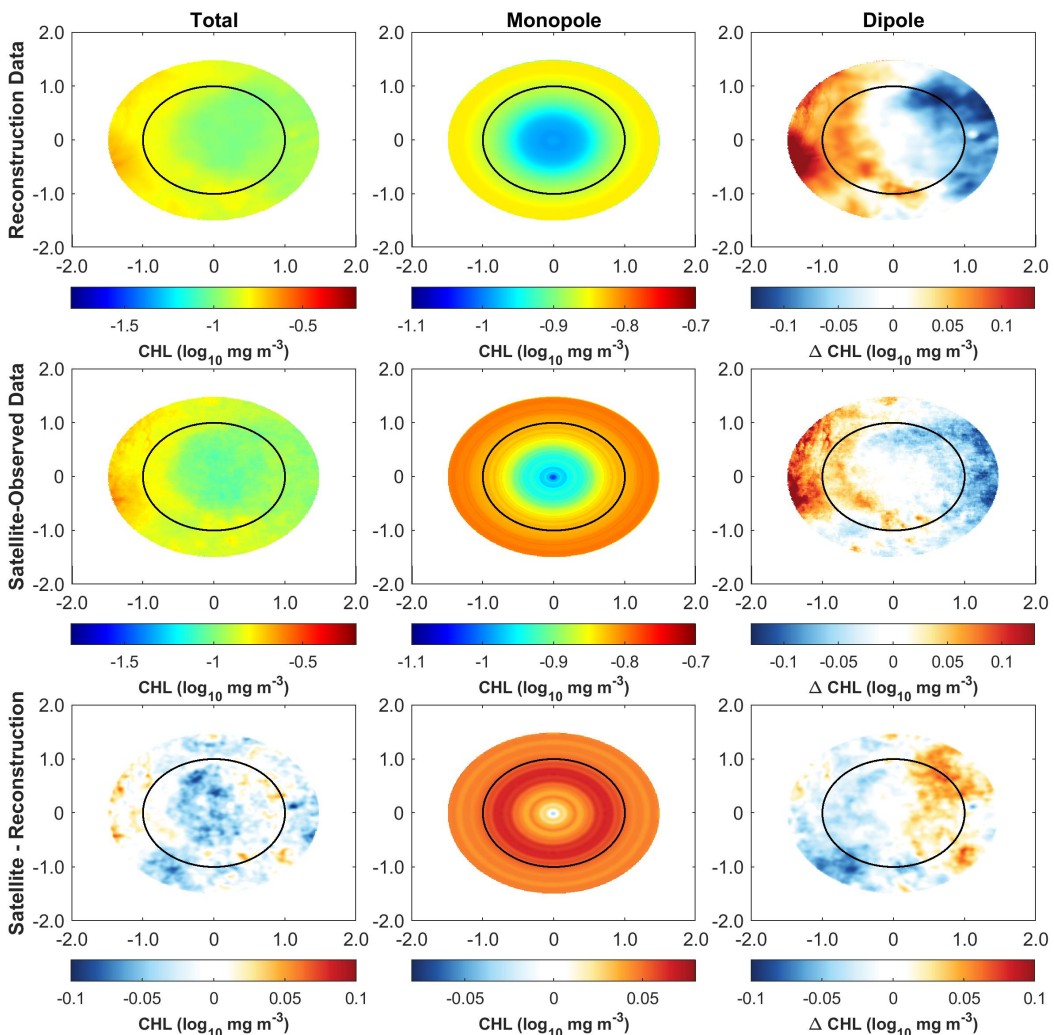

**Figure 19: Synthesized mean chlorophyll perturbation within the anticyclonic eddy (#27708868) and separation of the monopole chlorophyll perturbation and dipole chlorophyll perturbation (upper panel: computed from reconstructed data; middle panel: obtained from satellite observations; bottom panel: difference in chlorophyll perturbation computed from the two types of data)**

To further investigate the variations in chlorophyll levels related to eddy development, we utilized the Lagrangian method to scale the eddy life cycle from 0 to 1. We subsequently determined the chlorophyll perturbations for each 1/10th of the eddy's lifespan and analyzed the changes in the chlorophyll-a concentration in the radial direction (Fig. 20). During the initial phase of eddy formation (Normalized Day 1, ND1), the chlorophyll in the surface layer evenly spread. The difference in

chlorophyll concentration between the central edge and outer edge was approximately 0.01 $\log_{10}$ mg m$^{-3}$, with the chlorophyll-a concentration being slightly greater in the central edge than in the outer edge. During the strengthening stage (ND2-ND3), a phenomenon emerged where the chlorophyll-a

concentration in the center of the eddy was lower than that in the surrounding area, although a distinct low-chlorophyll core had not yet developed. During the mature stage, the eddy exhibited a central region with a relatively low concentration of chlorophyll, which was dispersed within a radius of 0-0.5 times the eddy's size. During the later phase of the maturation stage (ND6-ND8), chlorophyll ring structures form near the eddy's 1-fold radius where chlorophyll accumulates due to the influence of the anticyclone. This fact suggested that agitation at the edge became the primary mechanism for disrupting chlorophyll during the maturation period. During the extinction period, the chlorophyll-a concentration in the core was evenly distributed, and the chlorophyll-a concentration in the center of the edge was somewhat greater than that in the outer edge of the eddy. Once the eddy vanishes, the disturbance in chlorophyll likewise ceases.

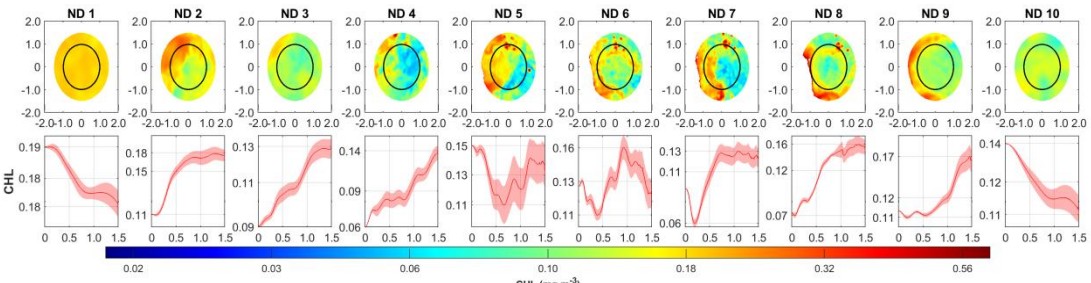

**Figure 20: Chlorophyll perturbations and radially averaged chlorophyll-a concentrations in anticyclonic eddy (#27708868) at different life stages**

### 5. Data availability

The reconstructed daily chlorophyll-a datasets has been published and is available at: https://doi.org/10.5281/zenodo.10478524 (Ye et al., 2024)

### 6. Conclusions

Our proposed OI-SwinUnet reconstruction model is better suited for reconstructing large-scale, high-resolution satellite-observed chlorophyll-a concentration products in the South China Sea than traditional reconstruction schemes such as DINEOF, OI, and Unet, which rely on convolutional operations. The analysis of the merged Aqua and Terra chlorophyll-a concentration products from 2013 to 2017 indicates that the OI-SwinUnet model outperforms the other models. The OI scheme is a spatially interpolated method that does not correlate information between products in the temporal

dimension. DINEOF incorporates time data while sacrificing a significant portion of the fine-grained spatial data. However, Unet is unable to handle a reconstruction task at such a wide scale and high resolution because of the intrinsic constraints of convolutional operations. These processes make it challenging for Unet to learn explicit correlations between global and long-range semantic information. The multihead attention mechanism and hierarchical window architecture of OI-SwinUnet effectively address the issue of long-range dependence and exhibit strong relationships with global information. With the inclusion of the OI module, the model can now incorporate in situ observational data alongside satellite data, resulting in a more dependable data product.

The reconstruction performance of OI-SwinUnet in various regions of the South China Sea is dependent on the extent of coverage provided by the satellite observation products. The reconstruction results will be more credible if the region is less impacted by weather conditions and has greater coverage of relevant data. The good generalization ability of OI-SwinUnet was proven by a designed experiment with different masking patterns and different masking rates.

The application potential of the reconstructed chlorophyll-a concentration product in the South China Sea is significant. The reconstructed dataset accurately represents both the seasonal-scale temporal and spatial patterns of sea-surface chlorophyll-a changes in the South China Sea and the rapid changes in marine phenomena at the weather scale. This includes capturing the impact of plume fronts on surface phytoplankton changes in the upwelling zone through nutrient input. The reconstructed data can be utilized not only for studying the primary ecological effects of mesoscale eddy activities in specific regions but also for illustrating the chlorophyll perturbations of each individual eddy at various life stages. This approach offers researchers a novel and comprehensive viewpoint on eddy studies. In the future, we can employ reconstructed data to further enhance marine scientific study. The real-time, large-scale, long time-series, and stable observation data obtained from satellite remote sensing are highly advantageous for monitoring and assessing ocean carbon fluxes and stocks, as well as studying the ocean carbon cycle. Additionally, these data serve as a motivating factor for advancing the application of remote sensing of ocean color in the study of the ocean carbon cycle. Currently, there are significant uncertainties and challenges in estimating the ocean carbon sink based on actual measurements. The OI-SwinUnet deep learning reconstruction model and high-precision remote sensing reconstruction products are crucial in studying the spatial and temporal distribution pattern of carbon parameters in response to global changes. They also help reduce uncertainties in estimating

carbon fluxes and stocks.

**Author contributions**

HY designed the research, developed the model and datasets. CY pre-processed the remote sensing data. YD processed the model validation. CC provided guidance on data processing. HY wrote the manuscript with feedback from all authors. All work was with the supervision of ST.

**Competing interests**

The contact author has declared that none of the authors has any competing interests.

**Acknowledgments**

The authors would like to thank the NASA Goddard Space Center for providing the MODIS data and the NASA OBPG group for providing the SeaDAS software package. The colleagues in the Ocean Color Group of the South China Sea Institute of Oceanology, Chinese Academy of Sciences, are greatly appreciated for their effort in collecting and processing the samples. The numerical analysis was

supported by the High Performance Computing Division of the South China Sea Institute of Oceanology.

**Financial support**

This research was funded by the Strategic Priority Research Program of the Chinese Academy of Sciences (No. XDA0370201), the Science and Technology Program of Guangzhou, China (No.

202201010101), the Guangdong Special Key Team Program (No. 2019BT02H594), the Research Fund of South China Sea Institute of Oceanography (No. SCSIO202209), the State Key Laboratory of Tropical Oceanography Independent Research Fund (No. LTOZZ2103), the Science and Technology Development Foundation of South China Sea Bureau, Ministry of Natural Resources (No. 230207), and State Key Laboratory of Tropical Oceanography, South China Sea Institute of Oceanography,

Chinese Academy of Science (No. LTO2313).

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
