# Peer review of "A daily reconstructed chlorophyll-a dataset in the South China Sea from MODIS using OI-SwinUnet"

_Earth System Science Data, 2024_

## Referee Comment (RC1)

In this manuscript, a deep learning model, OI-SwinUnet, is proposed for the reconstruction of remotely sensed chlorophyll products, and the model is used to generate MODIS chlorophyll-a concentration products in the South China Sea (SCS) from 2013 to 2017. The reconstructed products can be used to obtain comprehensive spatiotemporal continuum data in the SCS. Research on deep learning processing techniques for remote sensing data is currently quite popular. The proposed deep learning framework does a very creative job of addressing the crucial issue of missing data. The reconstructed data can be applied in ecological monitoring of small- and mesoscale processes in the ocean, indicating that the authors have accomplished excellent results. I think the paper satisfies the goals and specifications of this journal. Naturally, I have some particular comments that the authors should clarify or revise before the paper is officially accepted.

The first comments is about the input. Why did the author select anomalies for SwinUnet's inputs from the first and last 15 days, respectively? (To put it another way, could this last for three days or a week?)

Secondly, the author's method of demonstrating the model's reconstruction performance under various mask percent settings is commendable, but it appears that the graph's performance findings are not sufficiently clear (see from Fig. 12). Here, two recommendations are made: first, select an alternative reconstruction product at a time when there will be a sufficient difference to support the author's position; and second, include graphs with mask percentages of 30% and 70%, i.e., set the plot step size to 20% to reflect more specific information about the changes.

Thirdly, in my opinion, one of the best parts of this research is the use of reconstructed data in specific instances of mesoscale eddies. A useful database for researching the ecological effects of small- and mesoscale ocean phenomena may be produced if the chlorophyll data reconstructed using OI-SwinUnet, as suggested by the authors, are able to accurately restore the chlorophyll information of the missing regions.

Fourthly, the authors link upwelling to the high chlorophyll values seen along the Vietnamese coast throughout the summer. Have other studies verified that upwelling at

this location results in changes in chlorophyll, and can relevant literature be shown to bolster the authors' claims?

Other minor comments:

1. Is the "satellite-derived" in the x-axis of Fig.7 extracted from aqua or terra, or is the data merged from two sensors? please clarify this.

2. The better background color of Fig. 1, 2, 3, 4, 16, and 17 is white.

3. It is recommended that Fig. 10's colormap be changed to something other to make it more clear, such as jet.

4. Fig. 16a is rough and it should be improved. The below figure in Fig. 16a shows too little information.

---

## Referee Comment (RC2)

**Comments for the manuscript by Ye et al.:**

The manuscript "A daily reconstructed chlorophyll-a dataset in South China Sea from MODIS using OI-SwinUnet" by Haibin Ye et al. presents a novel approach to reconstructing missing chlorophyll-a data in the South China Sea using a combination of Optimal Interpolation (OI) and Swin Transformer-based Unet (SwinUnet). This method addresses the common issue of sporadic missing data in satellite-derived chlorophyll-a products, which can significantly hinder marine research and applications. By leveraging the strengths of both OI and deep learning through SwinUnet, the authors aim to provide a more accurate and spatially comprehensive dataset of chlorophyll-a concentrations. This reconstruction is crucial for understanding ecological effects in the South China Sea and improving the utilization of remote sensing data in marine studies.

**General comments:**
1. Innovation: The combination of OI and SwinUnet for reconstructing missing chlorophyll-a data is commendable. It addresses a significant gap in marine science research by providing a method to fill in missing satellite data, which is a common problem due to factors like cloud cover, sun glint, and sensor limitations.

2. Evaluation: The manuscript does an excellent job of comparing the OI-SwinUnet method against other common reconstruction methods such as DINEOF, OI, and Unet, demonstrating its superiority in handling missing data reconstruction in the South China Sea.

3. Applicability and impact: The study's findings have significant implications for marine science, particularly in understanding the spatial and temporal distribution of chlorophyll-a in the South China Sea. The reconstructed dataset can enhance studies related to marine ecology, biogeochemical cycles, and ocean dynamics.

**Specific comments:**
1. I would recommend revising the title of the manuscript by adding "the" before "South China Sea".

2. Introduction: The introduction provides a solid rationale for the study, situating it well within the current state of literature. However, it would benefit from a more detailed discussion of recent advances in data reconstruction techniques, particularly those employing machine learning and deep learning methods beyond the marine sciences, to highlight the novel contribution of OI-SwinUnet.

3. Method: The detailed explanation of the OI-SwinUnet model, including its components and the rationale behind its design, provides clarity and demonstrates the robustness of the approach. However, more details on the specific configurations of the SwinUnet architecture used in this study (e.g., number of layers, heads in multi-head

self-attention) could further enhance this section.

4. Results: The results convincingly demonstrate the superiority of the OI-SwinUnet method over traditional reconstruction methods like DINEOF, OI, and Unet through comprehensive statistical evaluation. While the statistical metrics employed are appropriate, incorporating a discussion on the practical significance of these statistical improvements in real-world applications would add value. Moreover, presenting case studies or specific instances where the reconstructed data reveal new insights about marine ecological processes could illustrate the method's impact more vividly.

4. Validation and metrics: The use of various statistical metrics (RMSD, $R^2$, bias) for model evaluation is appropriate. Additionally, assessing the model's performance across different missing data patterns (MCAR, MAR, MNAR) adds to the robustness of the findings. Future work could include comparisons with in-situ measurements if available, to further validate the reconstructed chlorophyll-a concentrations against ground truth data, and discussing how these advancements can influence our understanding of phytoplankton dynamics in response to climate change.

5. Discussion on limitations and future work: While the manuscript highlights the advantages of the OI-SwinUnet method, a more detailed discussion on its limitations and potentials for improvement would be valuable. For instance, how does the method perform in extremely turbid waters or under conditions of very high cloud cover? Also, exploring the potential of incorporating additional satellite sensors or data sources could be mentioned as a direction for future research.

6. Implication: The manuscript could benefit from a more detailed discussion on how the reconstructed dataset can be used to advance marine science research, beyond the examples provided. Potential applications in climate change studies, marine resource management, and oceanic carbon cycle research could be explored.

---

## Author Response (AR1)

Dear Reviewers:

Thank you for the comments concerning our manuscript entitled "A daily reconstructed chlorophyll-a dataset in South China Sea from MODIS using OI-SwinUnet" (ID: ESSD-2024-6). Those comments are all valuable and very helpful for revising and improving our paper, as well as the important guiding significance to our researches. We have studied comments carefully and have made correction which we hope meet with approval.

Revised portion are highlighted using the "Track Changes" function in the paper. The main corrections in the paper and the responds to the Reviewer's comments are as flowing:

Responds to the Reviewers' comments:

**Reviewer #1:**

**MAJOR COMMENTS**

1. **The first comments is about the input. Why did the author select anomalies for SwinUnet's inputs from the first and last 15 days, respectively? (To put it another way, could this last for three days or a week?)**

   Response: There are a large number of mesoscale processes such as eddies and fronts in the South China Sea, and their time scales range from a few days to several months. Theoretically, the longer the duration of the input variables, the more useful features the model can learn from the training. However, practically speaking, it is not possible to maximize the length of the input variables without any limitations, and the choice of fifteen days for the model inputs is a combination of many factors. Such a choice covers a complete mesoscale process as much as possible, while taking into account the computational efficiency. If three or seven days are used instead of fifteen days, the model will learn the features of the mesoscale process relatively poorly.

   In the revised manuscript, we have explain the reason for choosing 15 days of data before and after as input to the model (Lines: 180-186).

2. **Secondly, the author's method of demonstrating the model's reconstruction performance under various mask percent settings is commendable, but it appears that the graph's performance findings are not sufficiently clear (see from Fig. 12). Here, two recommendations are made: first, select an alternative reconstruction product at a time when there will be a sufficient difference to support the author's position; and second, include graphs with mask percentages of 30% and 70%, i.e., set the plot step size to 20% to reflect more specific information about the changes.**

   Response: Thanks for this useful comment. We have selected an image with more pronounced differences to show the specific performance of the model under the three different masking schemes. More detailed variations are shown for conditions where the mask percentage varies

from 10% to 90% (in steps of 20%). Since a new figure has been added to the revised manuscript, the original serial numbers of Fig. 12 and 13 have been changed to Fig. 13 and 14. Fig. 13 "Spatial distribution" and Fig. 14 "Performance" have been replaced due to the change in remote sensing imaging time. From the comparison of the old and new versions of these two figures, it can be seen that the large differences between the reconstruction results and satellite-derived values in the "spatial distribution" figure are mostly distributed in the near-shore area, and the changes in the statistical metrics in the "performance" figure have similar characteristics. This shows that our proposed reconstruction model has good robustness.

The old and new version of Fig. 13 and Fig. 14 are listed below:

[Figure]

3. **Thirdly, in my opinion, one of the best parts of this research is the use of reconstructed data in specific instances of mesoscale eddies. A useful database for researching the ecological effects of small- and mesoscale ocean phenomena may be produced if the chlorophyll data reconstructed using OI-SwinUnet, as suggested by the authors, are able to accurately restore the chlorophyll information of the missing regions.**

Response: Thanks for this comment. We hope to provide a spatiotemporally complete

chlorophyll-a concentration dataset that can satisfy the need for weather-scale observations and can accurately reveal the localized episodic occurrences of phytoplankton. From the results, the chlorophyll perturbations of eddies at different life stages can be depicted in more detail using reconstructed data compared to direct observations from satellites.

4. **Fourthly, the authors link upwelling to the high chlorophyll values seen along the Vietnamese coast throughout the summer. Have other studies verified that upwelling at this location results in changes in chlorophyll, and can relevant literature be shown to bolster the authors' claims?**

Response: Thanks for this comment. The upwelling along the eastern coast of Vietnam forms during May-September and reaches its mature stage during July-August (Fang et al. 2012; Voss et al. 2006). Past studies have shown that this summer upwelling is always in the form of a jet-like cold tongue (or cold patch) originating off the coast of Vietnam between 9° and 15° N (Hein et al. 2013). After generation, the cold tongue may extend eastward or northeastward into the central South China Sea (Gan et al. 2006). In the offshore region, the upwelling is usually accompanied by the Vietnamese cold eddy (Hu and Wang 2016). In addition to the cold water observed in upwelling regions, high chlorophyll-a concentrations are often reported (Ho et al. 2000; Li et al. 2014; Zhao and Tang 2007). By providing deep nutrient-rich water, upwelling stimulates the growth of phytoplankton in the euphotic zone, thus significantly altering the trophic state of the Vietnamese nearshore region (Bombar et al. 2010). Driven by transport in offshore currents, upwelling nutrients and stimulated high Chl-a ($\geq$0.2 mg m$^{-3}$) can extend from the coast to 116°E, creating 'Chl-a jets' (Chen, Xiu and Chai 2014).

We have added these content in the revised manuscript (Line: 485-497).

Newly Added References

Fang, G., Wang, G., Fang, Y., & Fang, W. (2012). A review on the South China Sea western boundary current. *Acta Oceanologica Sinica, 31*, 1-10

Voss, M., Bombar, D., Loick, N., & Dippner, J.W. (2006). Riverine influence on nitrogen fixation in the upwelling region off Vietnam, South China Sea. *Geophysical Research Letters, 33*, L07604

Hein, H., Hein, B., Pohlmann, T., & Long, B.H. (2013). Inter-annual variability of upwelling off the South-Vietnamese coast and its relation to nutrient dynamics. *Global and Planetary Change, 110*, 170-182

Gan, J., Li, H., Curchitser, E.N., & Haidvogel, D.B. (2006). Modeling South China Sea circulation: Response to seasonal forcing regimes. *Journal of Geophysical Research, 111*, C06034

Hu, J., & Wang, X.H. (2016). Progress on upwelling studies in the China seas. *Reviews of Geophysics, 54*, 653

Ho, C.R., Kuo, N.J., Zheng, Q., & Soong, Y.S. (2000). Dynamically active areas in the South China Sea detected from TOPEX/POSEIDON satellite altimeter data. *Remote Sensing of Environment, 71*, 320-328

Li, Y., Han, W., Wilkin, J.L., Zhang, W.G., Arango, H., & Zavala-Garay, J. (2014). Interannual variability of the surface summertime eastward jet in the South China Sea. *Journal of Geophysical Research: Ocean, 119*, 7205-7228

Zhao, H., & Tang, D.L. (2007). Effect of 1998 El Niño on the distribution of phytoplankton in the South China Sea. *Journal of Geophysical Research, 112*, C02017

Bombar, D., Dippner, J.W., Doan, H.N., Ngoc, L.N., Liskow, I., Loick-Wilde, N., & Voss, M. (2010). Sources of new nitrogen in the Vietnamese upwelling region of the South China Sea. *Journal of Geophysical Research, 115*

Chen, G., Xiu, P., & Chai, F. (2014). Physical and biological controls on the summer chlorophyll bloom to the east of Vietnam. *Journal of Oceanography, 70*, 323-328

**MINOR COMMENTS**

1. **Is the "satellite-derived" in the x-axis of Fig.7 extracted from aqua or terra, or is the data merged from two sensors? please clarify this.**

   Response: Thanks for this comment. The "satellite-derived" represents merged products from Aqua and Terra. We have clarify this information in Fig. 7's caption.

[Figure]

Figure 7: Scatter plots between satellite-derived (**merged products from Aqua and Terra**) and reconstructed chlorophyll-a concentration of different models, data acquired on (Upper panel: 11th February, 2014. Middle panel: February 27, 2015. Lower panel: January 5, 2016)

2. **The better background color of Fig. 1, 2, 3, 4, 16, and 17 is white.**

Response: Thanks for this comment. We have changed the background color to white.

3. **It is recommended that Fig. 10's colormap be changed to something other to make it more clear, such as jet.**

Response: Thanks for this comment. We have changed the colorbar to jet.

4. **Fig. 16a is rough and it should be improved. The below figure in Fig. 16a shows too little information.**

Response: Thanks for this comment. We have redrawn Fig. 17(a) (formerly Fig. 16(a)) in the hope of better demonstrating the interaction between plume front and upwelling.

[Figure]

Figure 16(a)

**Reviewer #2:**

**GENERAL COMMENTS**

5. **Innovation: The combination of OI and SwinUnet for reconstructing missing chlorophyll-a data is commendable. It addresses a significant gap in marine science research by providing a method to fill in missing satellite data, which is a common problem due to factors like cloud cover, sun glint, and sensor limitations.**

Response: Thanks for this comment. The lack of complete satellite observation data impedes

the utilization of satellite data in the domain of oceanic research. Deep learning offers significant potential in the realm of ocean remote sensing by extracting complex features from images using a vast quantity of data. We are developing a deep learning model using the SwinUnet framework to reconstruct sea surface chlorophyll data obtained from remote sensing. Our goal is to provide continuous and complete datasets that are accurate and reliable, which will be valuable for researchers.

6. **Evaluation: The manuscript does an excellent job of comparing the OI-SwinUnet method against other common reconstruction methods such as DINEOF, OI, and Unet, demonstrating its superiority in handling missing data reconstruction in the South China Sea.**

Response: Thanks for this comment. The current major remote sensing data reconstruction methods can be classified as traditional algorithms, including DINEOF and OI. Recently, researchers have also endeavored to do research on deep learning remote sensing reconstruction models, such as CNN-Unet. This work presents an analysis of the benefits and challenges encountered by DINEOF, OI, and CNN-Unet in the process of reconstructing remote sensing data. We also conduct a comparative evaluation of these methods with our suggested approach, OI-SwinUnet. The findings indicate that OI-SwinUnet is capable of preserving and restoring a greater amount of small- and meso-scale details compared to traditional algorithms. Furthermore, it is better suited than CNN-Unet for reconstructing extensive, high-resolution chlorophyll a concentration products detected by satellites in the South China Sea.

7. **Applicability and impact: The study's findings have significant implications for marine science, particularly in understanding the spatial and temporal distribution of chlorophyll-a in the South China Sea. The reconstructed dataset can enhance studies related to marine ecology, biogeochemical cycles, and ocean dynamics.**

Response: Thanks for this comment. It is our expectation that the rebuilt dataset will be useful not only for analyzing the temporal and spatial aspects of chlorophyll distribution at the surface of the South China Sea under long time periods, but also for capturing the phytoplankton outburst events that happen locally or by accident.   In the manuscript, we also analyze the differences between reconstructed datasets and satellite observation datasets in the application of monitoring the ecological effects of mesoscale eddies. Since the reconstructed dataset is characterized by spatiotemporal integrity, it is also particularly suitable for the changing pattern of surface chlorophyll over the full life cycle of eddies, which will help us to have a more comprehensive understanding of the ecological effects of mesoscale eddies.

**SPECIFIC COMMENTS**

5. **I would recommend revising the title of the manuscript by adding "the" before "South China Sea".**

Response: Thanks for this comment. We have revised the title by adding "the" before "South

China Sea".

6. **Introduction: The introduction provides a solid rationale for the study, situating it well within the current state of literature. However, it would benefit from a more detailed discussion of recent advances in data reconstruction techniques, particularly those employing machine learning and deep learning methods beyond the marine sciences, to highlight the novel contribution of OI-SwinUnet.**

Response: Thanks for this useful comments. In the revised manuscript, we discuss recent developments in two reconstruction approaches, OI and DINEOF (Line: 56-85).

"The OI algorithm leverages the conservative nature of marine elements and takes into account the spatial distribution characteristics of each element. It interpolates the unevenly distributed data to the corresponding grid points, resulting in an optimal estimation. This algorithm increases the coverage area and data density, allowing for the simultaneous use of observation data with varying error characteristics. It effectively addresses the issue of sparse spatial distribution of marine data. The optimal interpolation method has gained global recognition since the 1980s and has been adopted by the U.S. National Meteorological Center (NMC) and the European Centre for Medium-Range Weather Forecasts (ECMWF) for assimilation analysis and numerical prediction (Shaw 1986). The method is extensively employed in the marine domain to reconstruct historical datasets of sea surface temperature (SST), in situ measurements, and sea level anomaly (SLA) datasets. Currently, it is the most often used data assimilation method in the field of marine meteorology. The assumption made by "OI" is that the datasets are independent in terms of space and time. However, it fails to adequately consider the spatial and temporal correlation of the data. The suboptimal computational efficiency of the optimal interpolation approach is also a constraining factor in its implementation.

DINEOF is a data reconstruction technique that relies on the use of Empirical Orthogonal Function (EOF). It possesses the benefit of internal adaptive correlation without requiring any predetermined values for variables. The cross-correction set is implemented to facilitate the optimal reduction of truncation and estimation errors when constructing the EOF by accounting for default values. This method not only addresses missing data and eliminates noise from the data image, but also produces a dynamically adjusted image that accurately represents the overall condition of the data and its temporal development trend. This is achieved by utilizing the most significant modes obtained through optimal truncation (Alvera-Azarate, Barth and Rixen 2005). Due to the fact that the initial modes in the DINEOF method, which are derived from the entire target dataset decomposed by EOF, represent changes that occur over a period of more than six months, the reconstruction of multi-year time scale large data volume satellite remote sensing datasets using the DINEOF method focuses primarily on capturing temporal and spatial large-scale information. It disregards the small-scale information from a few local observation points. Therefore, using the interpolated target ocean dataset with missing measurements generated by the DINEOF method is not suitable for studying temporal small-scale processes, such as local weather-scale phenomena."

We also present recent developments in deep learning models that combine CNNs and attention gates for chlorophyll product reconstruction in nearshore marine environments (Line: 88-93).

"Unet is a compact convolutional neural network architecture that includes an encoder-decoder framework, which involves downsampling and upsampling operations. Additionally, Unet incorporates Attention Gates (AGs) inside its network structure. By training Unet with AGs, the background regions in the image are suppressed while the salient features in the data-missing regions are highlighted. This leads to an improvement in the sensitivity of the model and the accuracy of reconstruction."

In the final section of the Introduction, we address the issues encountered by current data reconstruction techniques and propose a novel OI-SwinUnet scheme (Line: 116-122).

"This paper aims to address the existing challenges and research gaps in traditional reconstruction methods and CNN-based reconstruction models. It focuses on studying the mechanism of chlorophyll in multi-scale spatio-temporal changes in the South China Sea (SCS), including weather-scale. To achieve effective filling of missing data in remotely sensed data products, we proposes a novel approach called the OI-SwinUnet method. This method combines the techniques of optimal interpolation (OI) and SwinUnet, and utilizes a multi-scale optimal interpolation, quadratic revision of transformer-based U-type coding and decoding network."

Newly Added References

Alvera-Azarate, A., Barth, A., & Rixen, M. (2005). Reconstruction of Incomplete Satelite SST Data Sets Using Empirical Orthogonal Functions: Application to the Adriatic Sea Surface Temperature. *Ocean Modelling, 9*, 325-346

Shaw, B.D. (1986). *Data Assimilation Scientific Documentation Research Manual*. England: European Centre for Medium-Range Weather Forecasts

**7. Method: The detailed explanation of the OI-SwinUnet model, including its components and the rationale behind its design, provides clarity and demonstrates the robustness of the approach. However, more details on the specific configurations of the SwinUnet architecture used in this study (e.g., number of layers, heads in multi-head self-attention) could further enhance this section.**

Response: Thanks for this useful comment. In the revised manuscript, we have rewritten the SwinUnet part of Method. It is described in three parts:: "SwinUnet framework - Swin Transformer block - (S)W-MSA module", and the detailed configurations of SwinUnet are added, such as the frequency of upsampling/downsampling, the size of the feature map for each stage, the size of the window of the (S ) window size of W-MSA, the length of the vector of tokens, and so on.

(1) "SwinUnet framework" (Line 233-259): The Unet architecture serves as the fundamental framework for SwinUnet. The model comprises four primary components, namely, the encoder, decoder, bottleneck, and skip connections (Figure 3). Given that the original SwinUnet model

requires 3 channels of input data, we encountered disparity because our preprocessed data comprised 66 channels. To address this discrepancy, we introduce an additional convolutional layer prior to the patch partition layer. This new layer serves the purpose of transforming the data from its original 66 channels to the required 3 channels. In order to transform the image into an embedding sequence, we divide the entire input tensor into patches of size $4 \times 4$ that do not overlap. These patches are then flattened in the direction of the channels. By employing this partitioning technique, the dimensions of the feature map transform from $[H, W, 3]$ to $[H/4, W/4, 48]$. Next, the linear embedding layer linearly transforms the feature dimension of each pixel from 48 to $C$. This results in a change in the shape of the feature map from $[H/4, W/4, 48]$ to $[H/4, W/4, C]$. Within the encoder, the patches are inputted into the Swin Transformer block to facilitate learning, while the feature size and resolution stay constant. Simultaneously, the patch merging layer will decrease the quantity of feature maps by a factor of 2 through downsampling, while doubling the feature dimension compared to its original size. This step will be iterated three times in the encoder. The symmetric decoder, which relies on the Swin Transformer block, serves as the counterpart to the encoder. The deep features that were recovered are enlarged in the decoder using a patch expanding layer. The patch expanding layer transforms the feature maps of adjacent dimensions into higher resolution feature maps ($2 \times$ up-sampling) and reduces the feature dimensions by half. In order to prevent the failure of convergence in a deep Swin Transformer block, the bottleneck is constructed using only two SW-MSA modules. This construction ensures that the feature size and resolution stay unchanged. Like UNet, skip connections are employed to integrate multiscale information from the encoder with up-sampled features. Shallow and deep features are linked together to reduce the loss of spatial information caused by downsampling. Ultimately, the feature map's resolution is increased four times by utilizing the final patch expanding layer, resulting in a restoration to the original input resolution. Afterwards, a linear projection layer is used to generate pixelwise predictions using the upsampled features.

(2) "Swin Transformer block" (Line 260-268): The fundamental element of SwinUnet is the Swin Transformer block (Figure 4). The construction of the Swin Transformer block is based on the concept of the shift window. The Swin Transformer block is composed of two normalization layers (LNs), a multihead self-attention module, residual connections, and a multilayer perceptron (MLP) layer with a GELU nonlinear activation function (Xiao et al. 2020). The use of the window-based multihead self-attention module (W-MSA) and the shift window-based multihead self-attention module (SW-MSA) is observed in two consecutive transformer blocks (Figure 4). The formula for the block can be represented as follows.

(3) "(S)W-MSA module" (Line 269-275): The W-MSA module initially partitions the feature map into several windows based on the specified $M \times M$ dimensions. It subsequently computes the self-attention within each window independently. Nevertheless, the W-MSA module lacks the capability to transfer information between windows. Therefore, it becomes imperative to implement SW-MSA, which relies on shifted windows, in order to address this limitation. The SW-MSA module, together with the W-MSA module in the Swin Transformer block, forms a two-tier structure through which information can be passed through neighboring windows.

(4) "configurations of SwinUnet" (Line 281-295): The configuration of SwinUnet is shown in

Table 1. The downsampling (upsampling) rate refers to the frequency at which upsampling and downsampling are carried out by the patch merging layer and patch expanding layer. After resampling, the output feature maps for each stage have heights and widths of $[560 \times 560, 280 \times 280, 140 \times 140, 70 \times 70]$ accordingly. The window size for performing MSA and SW-MSA operations is set to 7×7. As a result, each stage contains a total of [6400, 1600, 400, 100] windows. The hidden size refers to the length of the vector associated with each token, which represents the feature dimension of the feature map. Upon traversing the linear embedding layer, the feature dimension of the feature map in Unet's stage 1 is augmented to 96, and thereafter doubles in size in the following stages. The depth refers to the quantity of W-MSA and SW-MSA modules present in the Swin Transformer block. Specifically, in the first three stages, the Swin Transformer block is composed of a double layer structure consisting of one W-MSA module and one SW-MSA module. In stage 4, often known as the bottleneck, there is only one SW-MSA module. The MLP size refers to the number of nodes in the first fully-connected layer of the MLP module, which is four times the hidden size. The "heads" parameter represents the number of nodes in both the W-MSA and SW-MSA in the Swin Transformer block.

Table 1 Detailed architecture configurations of SwinUnet

|  | Downsampling/Upsampling Rate (Output Feature Map Size) | Window size | Window Numbers | Hidden Size | Depth | MLP Size | Heads |
|---|---|---|---|---|---|---|---|
| Stage 1 | 4 (560×560) | 7×7 | 6400 | 96 | 2 | 384 | 3 |
| Stage 2 | 8 (280×280) | 7×7 | 1600 | 192 | 2 | 768 | 6 |
| Stage 3 | 16 (140×140) | 7×7 | 400 | 384 | 2 | 1536 | 12 |
| Stage 4 | 32 (70×70) | 7×7 | 100 | 768 | 1 | 3072 | 24 |

**8. Results: The results convincingly demonstrate the superiority of the OI-SwinUnet method over traditional reconstruction methods like DINEOF, OI, and Unet through comprehensive statistical evaluation. While the statistical metrics employed are appropriate, incorporating a discussion on the practical significance of these statistical improvements in real-world applications would add value. Moreover, presenting case studies or specific instances where the reconstructed data reveal new insights about marine ecological processes could illustrate the method's impact more vividly.**

Response: Thanks for this useful comment. We compared the effectiveness of OI-SwinUnet and the other three methods in filling in the gaps in the time series using two typical pixels (highly turbid water and clean water) in the revised manuscript. The results demonstrated that OI-SwinUnet could reasonably reproduce the inter-annual and seasonal variation patterns of chlorophyll in both the highly turbid water and the clean water. (Line: 362-375)

"Two representative pixels were sampled from the Pearl River Estuary in the northern part of

the South China Sea and the central part of the South China Sea. These pixels were chosen to represent highly turbid water and clean water, respectively (red triangles in Figure 10). The purpose was to compare the performance of OI-SwinUnet and three other methods in terms of filling gaps in time series data. The results indicate that our proposed OI-SwinUnet demonstrates strong resilience to localized extremes, typically outliers. Within the clean water region, the OI-SwinUnet, DINEOF, and OI methods are capable of analyzing the dynamic patterns of the chlorophyll time series. However, the Unet method performs slightly less accurately, as it tends to underestimate chlorophyll values in most time intervals. This discrepancy is particularly evident in time intervals where satellite observations are consistently absent. In areas with high levels of turbid water, the OI-SwinUnet method performs similarly to the DINEOF method during periods of consecutive high chlorophyll levels. Figure 9 demonstrates that DINEOF is more successful in reconstructing high chlorophyll levels. This suggests that the method can effectively fill in the gaps in the time series data, allowing for reasonable patterns of interannual variation in chlorophyll-a to be observed."

[Figure]

**Figure 9: Gap-filled time series of two represented pixels (a) clear water, and (b) high turbid water using DINEOF , OI, Unet and OI-SwinUnet methods**

9. **Validation and metrics: The use of various statistical metrics (RMSD, R2, bias) for model evaluation is appropriate. Additionally, assessing the model's performance across different missing data patterns (MCAR, MAR, MNAR) adds to the robustness of the findings. Future work could include comparisons with in-situ measurements if available, to further validate the reconstructed chlorophyll-a concentrations against ground truth data, and discussing how these advancements can influence our understanding of phytoplankton dynamics in response to climate change.**

Response: Thanks for this comment. The purpose of designing experiments with various missing data patterns and rates is to assess how the performance of the reconstruction model is affected by growing data sparsity and the complexity of the missing scenario. The findings indicate that OI-SwinUnet is capable of learning the characteristics at various spatial scales, even when the satellite observation data is intentionally concealed. This is achieved by utilizing the knowledge from both before and after time, resulting in the precise reconstruction of the areas that were previously missing. In situ measurements are crucial for validating models by providing accurate data. Regrettably, we did not gather any in situ measurements of chlorophyll from 2013 to 2017. The reviewer's comments serve as a valuable reference for our future endeavors. In our upcoming work, we will diligently gather numerous field observations to further assess and authenticate the OI-SwinUnet model.

10. **Discussion on limitations and future work: While the manuscript highlights the advantages of the OI-SwinUnet method, a more detailed discussion on its limitations and potentials for improvement would be valuable. For instance, how does the method perform in extremely turbid waters or under conditions of very high cloud cover? Also, exploring the potential of incorporating additional satellite sensors or data sources could be mentioned as a direction for future research.**

Response: Thanks for this useful comment. Through the selection of various pixels and the analysis of the time series of reconstruction results from different methods, we have discovered that while OI-SwinUnet generally performs well, its reconstruction performance is slightly inferior to that of DINEOF when dealing with highly turbid and chlorophyll-rich waters. This is an important aspect to consider in our future work. Furthermore, we have contemplated integrating additional sensors or data sources, such as the chlorophyll products obtained from Sentinel-3A/B OLCI. The OI module may integrate data from several sensors to create a combined product from multiple satellites. This merged product is then passed to the SwinUnet module for reconstruction. Undoubtedly, there are still pending tasks that need to be completed prior to integrating additional satellite products, including doing study on the coherence of those products.

11. **Implication: The manuscript could benefit from a more detailed discussion on how the reconstructed dataset can be used to advance marine science research, beyond the examples provided. Potential applications in climate change studies, marine resource management, and oceanic carbon cycle research could be explored.**

Response: Thanks for this comment. We have included a potential application to the conclusion section such as the reconstruction of datasets for the purpose of advancing carbon cycle research in the final portion of the study.

Line 679-689: "In the future, we can employ reconstructed data to further enhance marine scientific study. The real-time, large-scale, long time-series, and stable observation data obtained from satellite remote sensing are highly advantageous for monitoring and assessing ocean carbon fluxes and stocks, as well as studying the ocean carbon cycle. Additionally, these

data serve as a motivating factor for advancing the application of remote sensing of ocean color in the study of the ocean carbon cycle. Currently, there are significant uncertainties and challenges in estimating the ocean carbon sink based on actual measurements. The OI-SwinUnet deep learning reconstruction model and high-precision remote sensing reconstruction products are crucial in studying the spatial and temporal distribution pattern of carbon parameters in response to global changes. They also help reduce uncertainties in estimating carbon fluxes and stocks."

We appreciate for Editors/Reviewers' warm work earnestly, and hope that the correction will meet with approval.

Once again, thank you very much for your comments and suggestions.

Yours sincerely,

Shilin Tang

Corresponding author:

Name: Shilin Tang

E-mail: sltang@scsio.ac.cn